# Achieving Hyperbolic-Like Expressiveness with Arbitrary Euclidean Regions: A New Approach to Hierarchical Embeddings

## Abstract

Hierarchical data is common in many domains like life sciences and e-commerce, and its embeddings often play a critical role. While hyperbolic embeddings offer a theoretically grounded approach to representing hierarchies in low-dimensional spaces, current methods often rely on specific geometric constructs as embedding candidates. This reliance limits their generalizability and makes it difficult to integrate with techniques that model semantic relationships beyond pure hierarchies, such as ontology embeddings. In this paper, we present RegD, a flexible Euclidean framework that supports the use of arbitrary geometric regions—such as boxes and balls—as embedding representations. Although RegD operates entirely in Euclidean space, we formally prove that it achieves hyperbolic-like expressiveness by incorporating a depth-based dissimilarity between regions, enabling it to emulate key properties of hyperbolic geometry, including exponential growth. Our empirical evaluation on diverse real-world datasets shows consistent performance gains over state-of-the-art methods and demonstrates RegD's potential for broader applications such as the ontology embedding task that goes beyond hierarchy.

## 1 Introduction

Embedding discrete data into low-dimensional vector spaces has become a cornerstone of modern machine learning. In Natural Language Processing (NLP), seminal works such as word2vec Mikolov et al. (2013) and GloVe Pennington et al. (2014) represent words as vectors to capture intricate linguistic relationships. Similarly, knowledge graph embedding methods Bordes et al. (2013); Sun et al. (2019); Balazevic et al. (2019) encode entities and relations as vectors with their semantics concerned to facilitate reasoning and prediction.

Our work focuses on embedding hierarchical data into low-dimensional spaces. Such data represents partial orders over sets of elements, denoted as $u \prec v$, where $u$ is a parent of $v$. These partial orders naturally manifest as trees or directed acyclic graphs (DAGs). The ability to effectively embed such hierarchical structures enables crucial operations like inferring sub- or superclasses of concepts and classifying nodes within graphs. These capabilities are essential for various tasks in knowledge management and discovery, particularly towards knowledge bases Shi et al. (2024); Abboud et al. (2020), ontologies He et al. (2024a); Chen et al. (2024) and taxonomies Shen & Han (2022).

Current methods for embedding hierarchical data can be broadly categorized into two paradigms: *region-based* and *hyperbolic* approaches. The region-based approach usually represents entities as geometric regions in the Euclidean space, capturing hierarchical relationships through intuitive region-inclusion. However, these methods often experience degraded performance in low-dimensional settings due to the crowding effect inherent in Euclidean spaces.

In contrast, the hyperbolic approach takes advantage of the unique geometric properties of hyperbolic spaces—specifically their exponential growth in distance and volume—enabling more effective embeddings of tree-structured data in low dimensions. However, as shown in Table 1, existing hyperbolic methods often rely on specialized constructed objects as embedding candidates Ganea et al. (2018b); Yu et al. (2024), limiting their generalizability to data that encodes richer semantics beyond hierarchy. For example, EntailmentCones Ganea et al. (2018b) and ShadowCones Yu et al. (2024) are not closed under intersection and thus cannot handle conjunction.

| Method | Embeddings of a node $u$ | Energy/score function for $u \prec v$ |
|---|---|---|
| **Poincaré** Nickel & Kiela (2017) | Points $\mathbf{u} \in \mathbb{H}^d$ | $(1 + \alpha(\|\mathbf{u}\| - \|\mathbf{v}\|))d_k(\mathbf{u}, \mathbf{v})$ |
| **EntailmentCones** Ganea et al. (2018b) | Cones with apex $\mathbf{u} \in \mathbb{H}^d$ and angle $\theta_{\mathbf{u}} = \arcsin(\frac{(1-\|\mathbf{u}\|^2)}{\|\mathbf{u}\|})$ | Angle-based function |
| **ShadowCones** Yu et al. (2024) | Cones with apex $\mathbf{u} \in \mathbb{H}^d$ and angle $\theta_{\mathbf{u}} = \arctan\sinh(\sqrt{k}r)$ | $d_{\mathbb{H}}(\mathbf{u}, \mathbf{v})$ **or** specific boundary distance |
| **RegD** (ours) | arbitrary regions in $\mathbb{R}^d$ (e.g., balls, boxes) | $d_{\text{bd}}(reg_u, reg_v) + \lambda \cdot d_{\text{dep}}(reg_u, reg_v)$ |

Table 1: Comparison with hyperbolic methods in $\mathbb{H}^d$ (curvature $-k$, distance $d_k$). $\alpha$, $r$ and $\lambda$ are predefined hyperparameters. Only the umbral half-space model of ShadowCones is shown, with boundary distance defined in Eq. 6.

In this paper, we propose a flexible framework named RegD for modeling hierarchical data by embedding arbitrary regions in Euclidean spaces. Our framework relies on two novel dissimilarity metrics, *depth dissimilarity* $d_{\text{dep}}$ and *boundary dissimilarity* $d_{\text{bd}}$, which combine the strengths of both region-based approaches in Euclidean spaces and hyperbolic methods. **The depth dissimilarity** (cf. Section 3.1) enables our model to achieve embedding expressiveness comparable to that of hyperbolic spaces by incorporating the "size" of the regions under consideration (cf. Theorem 1 and Propositions 1 and 3). **The boundary dissimilarity** (cf. Section 3.2) improves the representation of set-inclusion relationships among regions in Euclidean spaces. This allows for better identification of shallower and deeper descendants, thereby capturing hierarchical structures more effectively than traditional region-based approaches (cf. Proposition 2). Note that these dissimilarities may be negative or fail to satisfy the triangle inequality, and thus are generally not strict metric distances.

Notably, RegD can be applied to arbitrary regions, including common geometric representations such as balls and boxes. This generality enables broad applicability across diverse geometric embeddings for various tasks, extending beyond hierarchy alone data to ontologies that include hierarchies and more complex relationships in Description Logic Baader et al. (2017) (cf. Sections 4.2 and 4.3). Our main contributions are summarized as follows:

- We present a versatile framework that is able to embed hierarchical data as arbitrary regions in Euclidean space.
- We offer a rigorous theoretical analysis demonstrating that our framework retains the core embedding benefits of hyperbolic methods.
- Experiments on diverse real-world datasets demonstrate that our framework consistently outperforms existing approaches on embedding hierarchies and ontologies for reasoning and prediction.

For brevity, all proofs of theorems and properties are provided in Appendix A.

## 2  PRELIMINARIES AND RELATED WORKS

**Manifold and Hyperbolic Space**  A $d$-dimensional *manifold* Lee (2013), denoted $\mathcal{M}$, is a hypersurface embedded in an $n$-dimensional Euclidean space, $\mathbb{R}^n$, where $n \geq d$, and locally resembles $\mathbb{R}^d$. A *Riemannian manifold* $\mathcal{M}$ is a manifold equipped with a Riemannian metric, enabling the definition of a distance function $d_{\mathcal{M}}(\mathbf{x}, \mathbf{y})$ for $\mathbf{x}, \mathbf{y} \in \mathcal{M}$. *Hyperbolic space*, denoted $\mathbb{H}^n$, is a Riemannian manifold with a constant negative curvature of $-k$, where $k > 0$ Lee (2006). It can be represented using various isometric models, such as the Poincaré ball model (see Appendix C) or Poincaré half-space model, where the points are defined by the half-space: $U^n = \{\mathbf{x} \in \mathbb{R}^n : \mathbf{x}_n > 0\}$, and the hyperbolic distance between $\mathbf{x}, \mathbf{y} \in U^n$ is given by $d_k(\mathbf{x}, \mathbf{y}) = \frac{1}{\sqrt{k}} \operatorname{arcosh}\left(1 + \frac{\|\mathbf{x}-\mathbf{y}\|_2^2}{2\mathbf{x}_n \mathbf{y}_n}\right)$.

**Region-based Methods**  Region-based methods embed the nodes of a directed acyclic graph (DAG) into regions on Euclidean space or Riemannian manifolds, such as boxes Boratko et al. (2021); Zhang et al. (2022), balls Suzuki et al. (2019), and cones Vendrov et al. (2016), capturing hierarchical relationships through set-inclusion between these regions. Training is typically conducted using an inclusion loss, which is often defined in terms of the distance or volume of the regions Vendrov et al.

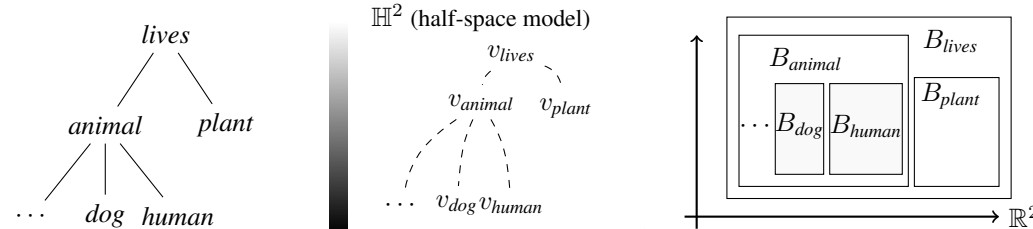

Figure 1: Demonstration of a taxonomy (left), its embeddings in the hyperbolic space (middle) and in the Euclidean space as boxes (right).

(2016). However, such loss functions may involve non-smooth operations, like maximization, which have been addressed through probabilistic Vilnis et al. (2018); Dasgupta et al. (2020) or smooth Boratko et al. (2021) approximations to improve performance. However, most of these methods suffer from the crowding effect in Euclidean space, which limits their expressiveness compared to our method and hyperbolic-based approaches.

**Hyperbolic Methods** Hyperbolic space embeddings Nickel & Kiela (2017); Sonthalia & Gilbert (2020); Sala et al. (2018) have been introduced to model hierarchical structures by leveraging favorable properties of hyperbolic space, such as its exponential growth. However, these methods primarily rely on pointwise distances to encode hierarchies, which is less intuitive for representing transitivity properties. In contrast, *EntailmentCones* Ganea et al. (2018b), and *ShadowCones* propose using regions in hyperbolic space that capture hierarchies through set inclusion, which can be regarded as region-based methods in hyperbolic spaces. Specifically, *EntailmentCones* introduces closed-form hyperbolic cones, defined by their apex coordinates, while *ShadowCones* use different cones are inspired by the physical interplay of light and shadow. However, as shown in Table 1, these regions all take specific forms, limiting their generalizability to other tasks or methods.

## 3 METHOD

### 3.1 DEPTH DISSIMILARITY: A SIMILARITY MEASURE INCORPORATING DEPTH

Unlike the Euclidean space, which is constrained by crowding effects that limit its embedding capacity, the hyperbolic space leverages exponential growth in distance and volume to offer superior embedding capabilities. This property makes the hyperbolic space particularly effective for representing tree-structured data in low-dimensional spaces. Notably, two key distinctions arise between region-based embeddings in the Euclidean space and hyperbolic embeddings:

1. *The hyperbolic space better discriminates different hierarchical layers than the Euclidean space.*
   Consider the taxonomy illustrated in Figure 1 (left). Intuitively, since *human* is a subcategory of *animal*, the semantic difference between *human* and *plant* should be greater than that between *animal* and *plant*. The hyperbolic space effectively captures this hierarchical relationship by permitting $d(v_{human}, v_{plant}) > d(v_{animal}, v_{plant}) + \Delta$, where $\Delta$ is an arbitrary gap. This arises from the property that the distance metric diverges to infinity near the boundary, as illustrated by the shadow dense of the bar on the left-hand side of $\mathbb{H}^2$ in Figure 1 (middle). In contrast, region-based embeddings on Euclidean space may violate this hierarchical constraint, potentially placing the box $B_{human}$ close to $B_{plant}$, resulting in a distance (*e.g.*, the Euclidean distance between box centers) similar to or even smaller than that between $B_{animal}$ and $B_{plant}$.

2. *The hyperbolic space enables the distinct representation of an arbitrary number of child nodes.*
   As demonstrated in Figure 1, the box embeddings in the Euclidean space face inherent limitations when representing multiple children of a node, such as *animal*. As the number of children increases, their corresponding boxes must cluster within $B_{animal}$, leading to crowding. In contrast, the hyperbolic space can accommodate an arbitrary number of children while maintaining distinct separations between them. This capability arises from the exponential growth of distance near the boundary of the hyperbolic space, which allows unlimited child nodes to be positioned distinctly

by placing them progressively closer to the boundary while preserving meaningful distances between them.

In the following sections, we introduce the notion of *depth dissimilarity* for regions in Euclidean space, which explicitly accounts for their "size." We show that this measure retains advantages analogous to those of hyperbolic spaces (Theorem 1), while offering a simpler structure that facilitates implementation (e.g., avoiding specialized techniques such as double-precision tensors and Riemannian-specific optimizers Bécigneul & Ganea (2019)) and enabling potential extensions (e.g., integration with ontology embeddings; see Section 4.2). Moreover, we establish that depth dissimilarity encompasses hyperbolic distance as a special case (Proposition 1), while providing greater flexibility by preserving key properties of hyperbolic geometry through simple polynomial functions (Proposition 2).

**Construction**   The depth dissimilarity serves as a similarity measure that quantifies the relationship between objects considering their hierarchical depth. As we use set-inclusion relations to model the hierarchy, this hierarchical depth can be represented through the size of the regions, such as their volumes or diameters. Formally, the depth dissimilarity is defined as follows, where the size of the regions is represented by a function $f(reg)$:

**Definition 1** (Depth Dissimilarity). *Let $\mathcal{R}$ be a collection of* parameterized regions[1] *in the $n$-dimensional Euclidean space $\mathbb{R}^n$, i.e., each region $reg \in \mathcal{R}$ is characterized by an $m$-dim parameter $P(reg) \in \mathbb{R}^m$. The depth dissimilarity between two regions $reg_1, reg_2 \in \mathcal{R}$, is defined as:*

$$d_{dep}(reg_1, reg_2) = g\left(\frac{\|P(reg_1) - P(reg_2)\|_p^p}{f(reg_1)f(reg_2)}\right) \qquad (1)$$

*where $\|\cdot\|_p$ is the p-norm (i.e., $\|\mathbf{x}\|_p = (\sum_i \mathbf{x}_i^p)^{1/p}$), and:*

- $g : \mathbb{R}_{\geq 0} \to \mathbb{R}_{\geq 0}$ *is an increasing function such that $g(x) = 0$ if and only if $x = 0$,*

- $f : \mathcal{R} \to \mathbb{R}_{>0}$ *is a function that measures the size of regions. It satisfies: $\lim_{reg \to \emptyset} f(reg) = 0$[2].*

We require $f$ and $g$ to have non-negative values to ensure the depth dissimilarity is non-negative. Additionally, we stipulate that $\lim_{reg \to \emptyset} f(reg) = 0$ to guarantee that as a region shrinks to an empty set, the dissimilarity between this object and others can approach infinity. This setting emulates the beneficial properties of the hyperbolic space, where the dissimilarity between two points can grow rapidly as they approach the boundary of the space (i.e., $\mathbf{x}_n = 0$ in the Poincaré half-space model). In our context, the boundary of the space $\mathcal{R}$ of a collection of (parametrized) regions in the Euclidean space is the empty set. By selecting an appropriate function $f$, we can control the rate at which the dissimilarity between two objects increases as they approach this boundary.

For the remainder of this paper, we restrict our theoretical analysis and implementations to boxes and balls as representative region types for clarity and tractability. It is important to note, however, that the theoretical results established herein naturally extend to other parameterized region representations, with analogous proof techniques applicable. For details, please refer to Appendix A.5.

**Example 1.** *Let $ball(\mathbf{c}, r) = \{\mathbf{x} \mid \|\mathbf{x} - \mathbf{c}\| \leq r\}$ be a* ball *defined by a center $\mathbf{c} \in \mathbb{R}^n$ and a radius $r > 0$. By setting $g(x) = x$ and $f(ball) = r$, we obtain a depth dissimilarity of the form:*

$$d_{dep}(ball_1(\mathbf{c}_1, r_1), ball_2(\mathbf{c}_2, r_2)) = \frac{\|\mathbf{c}_1 - \mathbf{c}_2\|_p^p + |r_1 - r_2|^p}{r_1 \cdot r_2}. \qquad (2)$$

**Example 2.** *Let $box(\mathbf{c}, \mathbf{o}) = \{\mathbf{x} \in \mathbb{R}^n \mid \mathbf{c} - \mathbf{o} \leq \mathbf{x} \leq \mathbf{c} + \mathbf{o}\}$ be a* box *defined by a center $\mathbf{c} \in \mathbb{R}^n$ and an offset $\mathbf{o} \in \mathbb{R}_{>0}^n$. By setting $g(x) = x$ and $f(box) = \|\mathbf{o}\|$, we obtain a depth dissimilarity:*

$$d_{dep}(box_1(\mathbf{c}_1, \mathbf{o}_1), box_2(\mathbf{c}_2, \mathbf{o}_2)) = \frac{\|\mathbf{c}_1 - \mathbf{c}_2\|_p^p + \|\mathbf{o}_1 - \mathbf{o}_2\|_p^p}{\|\mathbf{o}_1\| \cdot \|\mathbf{o}_2\|}. \qquad (3)$$

The following result highlights that the depth dissimilarity exhibits properties analogous to those of the hyperbolic distance discussed earlier in this section. Specifically, the depth dissimilarity **(1)**

---

[1]See Appendix A.5 for a formal definition. Examples as boxes and balls are shown in Example 1, 2.

[2]$reg \to \emptyset$ indicates that the region *reg* shrinks to the empty set (*e.g.*, its volume or diameter tends to zero).

effectively distinguishes concepts across different layers of the hierarchy with an arbitrary separation gap, denoted by $\Delta$ in item 1 of the following Theorem 1, and **(2)** distinctly represents an arbitrary number $n$ of children within a single parent by a dissimilarity greater than $M$, as demonstrated in item 2 of the theorem.

**Theorem 1.** *Consider the region space $\mathcal{B}^n$ consisting of **balls** in $\mathbb{R}^n$, with the depth dissimilarity defined in Example 1. The following properties hold:*

1. *For any $ball_1, ball_2 \in \mathcal{B}^n$ and any $\Delta > 0$, there exists a positive constant $r_0$ such that for any $ball(\mathbf{c}', r') \subseteq ball_2$, if $r' \leq r_0$, then $d_{dep}(ball_1, ball(\mathbf{c}', r')) > d_{dep}(ball_1, ball_2) + \Delta$.*

2. *For any $ball \in \mathcal{B}^n$, any integer $n$ and any $M > 0$, there exist subsets $ball_1, \ldots, ball_n \subseteq ball$ such that for any distinct $i, j \in \{1, \ldots, n\}$, we have $d_{dep}(ball_i, ball_j) > M$.*

*The same conclusions hold for boxes, where $r_0 \in \mathbb{R}_{>0}$ is replaced with $\mathbf{o}_0 \in (\mathbb{R}_{>0})^n$, and the condition $r' \leq r_0$ is replaced by $\mathbf{o}' \leq \mathbf{o}_0$ element-wise.*

**Comparison with the Hyperbolic Distance**   The following theorem shows that our depth dissimilarity encompasses hyperbolic distance as a special case, obtained with specific choices of functions $f, g$ in Definition 1.

**Proposition 1.** *Let $\mathbb{H}^{n+1}$ be the hyperbolic space with curvature $-1$. Assume the ball space $\mathcal{B}^n$ is parameterized as in Example 1, and equipped with the depth dissimilarity defined in Equation (1). Then the map $F: \mathcal{B}^n \to \mathbb{H}^{n+1}$ defined by $F(ball(\mathbf{c}, r)) = [\mathbf{c} : r]$ is an bijective isometry when $p = 2$, $g(x) = \text{arcosh}(x + 1)$, and $f(ball(\mathbf{c}, r)) = \sqrt{2}\, r$.*

Moreover, the following result demonstrates that even with a simple linear function $g(\cdot)$, our depth dissimilarity retains the ability to capture the hyperbolic structure. Specifically, there exists a map from the region space to the hyperbolic space that preserves the order of hyperbolic distances for all pairs of points, as stated below.

**Proposition 2.** *Following Proposition 1, let the depth dissimilarity $d_{dep}$ be redefined by replacing $g$ to $g(x) = k \cdot x \, (k > 0)$. Then the map $F$ retains the following monotonicity property: for any points $\mathbf{x}_1, \mathbf{x}_2, \mathbf{x}_3, \mathbf{x}_4 \in \mathbb{H}^n$, $d_{\mathbb{H}^n}(\mathbf{x}_1, \mathbf{x}_2) < d_{\mathbb{H}^n}(\mathbf{x}_3, \mathbf{x}_4)$ if and only if*

$$d_{dep}(F^{-1}(\mathbf{x}_1), F^{-1}(\mathbf{x}_2)) < d_{dep}(F^{-1}(\mathbf{x}_3), F^{-1}(\mathbf{x}_4)).$$

It is important to recognize that evaluation metrics such as F1 scores and Hits@k rely solely on the ranking induced by the scoring function. As a result, only the relative order of the scores matters, while their absolute values are less important. Therefore, preserving the same ranking as that produced by hyperbolic distances is sufficient to attain comparable performance. Consequently, the depth-based dissimilarity defined in Equations 2 and 3, with the function $g(x) = x$, are well-suited for our implementation, as justified by Proposition 3.

## 3.2   Boundary Dissimilarity: A Non-Symmetric Measure of Inclusion

Although the depth dissimilarity $d_{\text{dep}}$ introduced above has been shown to have great power for embedding hierarchical data, it is a symmetric metric and therefore inadequate for fully capturing the inherently non-symmetric hierarchical relationships between objects. To address this limitation, we introduce the boundary dissimilarity, specifically designed to reflect the partial order of regions defined by set inclusion.

Our boundary dissimilarity generalizes the signed-distance-to-boundary in ShadowCone (Theorem 4.2, Yu et al. (2024)) by extending its applicability from specialized hyperbolic cone geometries to arbitrary Euclidean regions. This generalization is formally defined in Definition 2. As a result, our boundary dissimilarity can be computed in a much simpler form (Example 3) compared to the formulation used in ShadowCone (Equation 6).

**Construction**   The boundary dissimilarity is defined to measure the minimal cost associated with transforming the spatial relationship between two regions $reg_1$ and $reg_2$. Specifically, it quantifies the cost of moving $reg_2$ out of $reg_1$ when $reg_2 \subseteq reg_1$, or moving $reg_2$ into $reg_1$ otherwise (when $reg_2 \nsubseteq reg_1$). This cost can be defined for arbitrary geometric objects based on either distance or

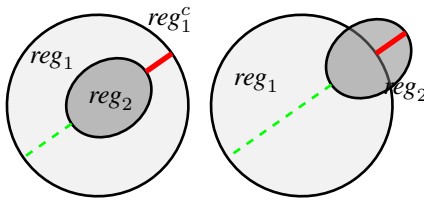 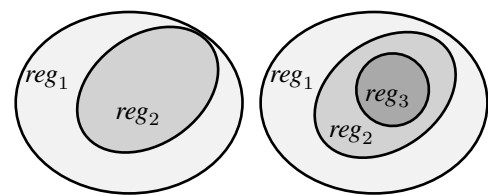

Figure 2: Illustration of $d_{\mathrm{bd}}(reg_1, reg_2)$ (red) for $reg_2 \subseteq reg_1$ (left) or $reg_2 \not\subseteq reg_1$ (right). Green lines shows the inverse: $d_{\mathrm{bd}}(reg_2, reg_1)$.

Figure 3: Illustration of internally tangent of item 1 (left) and item 2 (right) in Proposition 3.

volume within Euclidean or other spaces. Below, we introduce a boundary dissimilarity based on the Euclidean distance for two regions $reg_1, reg_2 \subseteq \mathbb{R}^n$, which consists of two cases:

1. *Containment ($reg_2 \subseteq reg_1$):* As illustrated in the left of Figure 2, when $reg_2$ is fully contained within $reg_1$, the boundary dissimilarity is defined by the minimum Euclidean distance between the complementary region $reg_1^c$ and the points in $reg_2$ (*i.e.*, length of the red line). This distance quantifies the minimum translation cost required to move at least a part of $reg_2$ out of $reg_1$.

2. *Non-Containment ($reg_2 \not\subseteq reg_1$):* As shown in the right of Figure 2, when $reg_2$ is not fully contained within $reg_1$, the boundary dissimilarity is defined as the maximum Euclidean distance from the points in $reg_2 \setminus reg_1$ to $reg_1$ (*i.e.*, length of the red line). This distance quantifies the minimum translation cost for moving $reg_2$ into $reg_1$.

Let $d(reg, \mathbf{x}) := \min\{||\mathbf{x} - \mathbf{y}||_2 \mid \mathbf{y} \in reg\}$ be the distance of a points $\mathbf{x}$ to a region $reg$ defined by the minimal distance from $\mathbf{x}$ to $\mathbf{y} \in reg$. The formal definition of the boundary dissimilarity is as follows:

**Definition 2** (Boundary Dissimilarity). *Given a region space $\mathcal{R}$, we define the boundary dissimilarity over $reg_1, reg_2 \in \mathcal{R}$ by:*

$$d_{bd}(reg_1, reg_2) = \begin{cases} -\min_{\mathbf{x}_2 \in reg_2} \{d(reg_1^c, \mathbf{x}_2)\} & \text{if } reg_2 \subseteq reg_1 \\ \max_{\mathbf{x}_2 \in reg_2 \setminus reg_1} \{d(reg_1, \mathbf{x}_2)\} & \text{else .} \end{cases} \tag{4}$$

Note that a negative sign is added to $d_{\mathrm{bd}}(reg_1, reg_2)$ in the containment case ($reg_2 \subseteq reg_1$) to clearly distinguish it from other cases. Moreover, the boundary dissimilarity is inherently asymmetric, that is, $d_{\mathrm{bd}}(reg_1, reg_2) \neq d_{\mathrm{bd}}(reg_2, reg_1)$ in general. For example, as illustrated in Figure 2, the boundary dissimilarity in the reverse order, $d_{\mathrm{bd}}(reg_2, reg_1)$, corresponds to the length of the green dashed line, which differs from the red line representing $d_{\mathrm{bd}}(reg_1, reg_2)$.

Moreover, for widely used geometric objects such as balls and boxes, the boundary dissimilarity can be computed efficiently using simple arithmetic operations.

**Example 3.** *If $reg_1 = ball(\mathbf{c}_1, r_1)$ and $reg_2 = ball(\mathbf{c}_2, r_2)$, the boundary dissimilarity have the form (here the two cases can be unified into a single formula):*

$$d_{bd}(reg_1, reg_2) = ||\mathbf{c}_1 - \mathbf{c}_2||_2 + r_2 - r_1$$

*For the case of boxes, where $reg_1 = box(\mathbf{c}_1, \mathbf{o}_1)$ and $reg_2 = box(\mathbf{c}_2, \mathbf{o}_2)$, the boundary dissimilarity have the form:*

$$d_{bd}(reg_1, reg_2) = \begin{cases} \max(|\mathbf{c}_1 - \mathbf{c}_2| + \mathbf{o}_2 - \mathbf{o}_1) & \text{if } reg_2 \subseteq reg_1, \\ ||\max\{|\mathbf{c}_1 - \mathbf{c}_2| + \mathbf{o}_2 - \mathbf{o}_1, \mathbf{0}\}||_2 & \text{else.} \end{cases}$$

*Here, $\max(\cdot)$ in the first line denotes the maximal value along all dimensions, while $\max\{\cdot, \cdot\}$ in the second line applies element-wise to the two vectors or values.*

The following proposition demonstrates that our definition of the boundary dissimilarity effectively captures the inclusion relationship between two regions in two key aspects, as illustrated in Figure 3: (i) it identifies whether one region is (exactly) contained within another, and (ii) it enhances

discrimination in inclusion chains, as smaller regions tend to have larger boundary dissimilaritys. This property is useful for distinguishing shallow children from deeper ones. It is worth noting that the proposition below applies to any regions, not only boxes or balls.

**Proposition 3.** *For the boundary dissimilarity $d_{bd}$ in Definition 2, the following properties hold:*

1. $d_{bd}(reg_1, reg_2) \leq 0$ *if and only if* $reg_1 \subseteq reg_2$. *Moreover,* $d_{bd}(reg_1, reg_2) = 0$ *if and only if* $reg_1$ *is internally tangent to* $reg_2$. *That is,* $reg_1 \subseteq reg_2$, *and their boundaries intersect at some point.*

2. *If* $reg_3 \subseteq reg_2 \subseteq reg_1$, *then* $d_{bd}(reg_1, reg_3) \leq d_{bd}(reg_1, reg_2)$.

**Specific Constructions for Boxes or Balls**   For specific geometric regions like balls and boxes, we can create specialized distance functions to measure the set-inclusion relationship based on their intrinsic geometric properties or established methods. Our framework accommodates these specialized metrics by allowing them to replace the general boundary dissimilarity function.

1. *Volume-based dissimilarity for boxes:* Since the volume of a box can be computed as the product of its offsets along different dimensions, we can define a partial distance based on volume:
$$d_{\text{vol}}(reg_1, reg_2) = -\ln\left(\frac{\text{vol}(reg_1 \cap reg_2)}{\text{vol}(reg_2)}\right). \tag{5}$$

2. *Hyperbolic dissimilarity for balls:* Yu et al. (2024) introduced a series of circular cones in hyperbolic space and defined a boundary dissimilarity based on the hyperbolic distance between the apex of these cones. By utilizing the natural mapping from balls to circular cones, we can derive a new boundary dissimilarity for balls as follows (see Appendix B for more details):
$$d_{\text{bd}}^{\text{cone}}(reg_1, reg_2) = \text{arcsinh}\left(\frac{\|\mathbf{c}_1 - \mathbf{c}_2\|_2 - r_1}{r_2}\right) + \text{arcsinh}(1) \tag{6}$$

### 3.3   Training

For a given pair $(u, v)$, we define their energy as a weighted sum with weight $\lambda$ that balances the contributions of the hyperbolic-like depth dissimilarity:
$$E(u, v) = d_{\text{bd}}(reg_u, reg_v) + \lambda \cdot d_{\text{dep}}(reg_u, reg_v), \tag{7}$$
We say that $u$ is considered a parent of $v$ (i.e., $u \prec v$) if $E(u, v) \leq t$, where $t$ is a threshold that achieved the best performance on the evaluation set.

For model training, we use $d_{\text{dep}}$ from Equation 2 or 3 with the contrastive loss from Yu et al. (2024):
$$\mathcal{L}(\gamma_1, \gamma_2) = \sum_{(u,v) \in P} \left( \max\{E(u, v), \gamma_1\} + \log\left( \sum_{(u,v') \in N} e^{\max\{\gamma_2 - d_{\text{bd}}(reg_u, reg_{v'}), 0\}} \right) \right), \tag{8}$$
where $P$ and $N$ denote positive and negative sample pairs, respectively. For **positive pairs** $(u, v) \in P$, the loss based on $E(u, v)$ promotes both the containment of $reg_v$ within $reg_u$ (via the $d_{\text{bd}}$ term) and their geometric similarity (via the $d_{\text{dep}}$ term), whose contributions are controlled by the weight $\lambda$ and the threshold $\gamma_1$. For **negative pairs** $(u, v') \in N$, since our primary goal is to push $reg_{v'}$ outside of $reg_u$, it is sufficient to use the boundary dissimilarity $d_{\text{bd}}(reg_u, reg_{v'})$ rather than the energy $E(u, v')$. A threshold $\gamma_2$ is used to regulate how far $reg_{v'}$ is pushed from $reg_u$.

## 4   Evaluation

Our experiments aim to address two questions: **(1)** How effectively do our methods capture hierarchical relationships? and **(2)** Can they generalize to tasks involving more than hierarchies?

We evaluate hierarchical relationship modeling using transitive DAGs (Section 4.1). To assess generalization, we test on *ontologies* (see Appendix E for a formal definition), which extend pure hierarchies by incorporating logical operations like conjunction ($\sqcap$) and existential quantifiers ($\exists r.$). Ontologies contain "SubclassOf" ($\sqsubseteq$) as a fundamental and ubiquitous relation, representing hierarchical structures. This makes ontologies ideal for evaluating beyond pure hierarchy modeling. Specifically, ontologies enable testing of: **(a)** Complex inferences tasks beyond transitive closure (Section 4.2); and **(b)** Link prediction across different, usually non-SubclassOf relations (Section 4.3). Due to space limitations, additional results and detailed experimental settings are provided in Appendix D.

Table 2: F1 score (%) on Mammal, WordNet noun, MCG, and Hearst. Results with * are coming from Yu et al. (2024).

| Method | | Mammal | | Noun | | MCG | | Hearst | |
|---|---|---|---|---|---|---|---|---|---|
| | | d=2 | d=5 | d=5 | d=10 | d=5 | d=10 | d=5 | d=10 |
| $\tau$Box | | 29.0 | 33.5 | 30.5 | 31.5 | 43.9 | 50.3 | 39.7 | 43.7 |
| OE | | 25.4 | 31.0 | 28.8 | 30.8 | 36.3 | 46.6 | 34.6 | 40.7 |
| ELBE (box baseline) | | 30.3 | 36.8 | 30.7 | 31.8 | 48.4 | 55.5 | 41.6 | 46.8 |
| ELEM (ball baseline) | | 27.7 | 28.8 | 28.6 | 29.5 | 35.7 | 38.6 | 34.6 | 36.7 |
| EntailmentCone* | | 54.4 | 56.3 | 29.2 | 32.1 | 25.3 | 25.5 | 22.6 | 23.7 |
| ShadowCone* | (Umbral-half) | 57.7 | 69.4 | 45.2 | 52.2 | 36.8 | 40.1 | 32.8 | 32.6 |
| | (Penumbral-half) | 52.8 | 67.8 | 44.6 | 51.7 | 35.0 | 37.6 | 26.8 | 28.4 |
| RegD | (box) | **64.9** | 71.6 | 53.8 | 51.3 | **50.7** | **58.5** | **42.8** | **49.6** |
| | (ball) | 62.7 | **71.8** | **58.4** | **59.1** | 44.9 | 46.8 | 37.7 | 37.7 |

## 4.1 Inferences over DAG

**Benchmark** Following Yu et al. (2024), we evaluate our method on four real-world datasets consisting of Is-A relations: MCG Wang et al. (2015); Wu et al. (2012), Hearst patterns Hearst (1992), the WordNet Fellbaum (1998) noun taxonomy, and its mammal subgraph. All models are trained exclusively on *basic edges*, which are edges not implied transitively by other edges in the graph. For validation and testing, we use the same sets as in Yu et al. (2024), consisting of 5% of *non-basic (inferred)* edges, ensuring a fair comparison. The hyperparameter settings are provided in Appendix D.1. We exclude non-basic edges from training since they can be transitively derived from basic edges. Including them would artificially inflate performance metrics without properly evaluating the embeddings' ability to capture hierarchical structures. For completeness, results for non-basic cases are provided in Appendix D.3.

**Baselines** We compare our method RegD with (i) hyperbolic approaches such as EntailmentCone Ganea et al. (2018b) and ShadowCone Yu et al. (2024), which is the latest method with the state-of-the-art performance; (ii) region-based methods like OrderEmbedding Bordes et al. (2013), and tBox Boratko et al. (2021). We also compare with the ontology embedding methods, ELBE Peng et al. (2022) and ELEM Kulmanov et al. (2019), which can be considered as the baseline approaches embedding the DAG as boxes or balls, respectively. We use F1-scores as in previous studies.

**Results** The performance comparison across different DAGs is shown in Table 2. RegD achieved the best performance on all four datasets. Notably, the box variant consistently outperformed the ball variant in most cases, which might be because boxes contain more parameters than balls when embedded in the same dimensional space. Interestingly, region-based methods outperformed hyperbolic methods on the MCG and Hearst datasets. However, on the Noun and mammal dataset, hyperbolic methods performed better. Nevertheless, our method performed consistently well in both cases, as it can adjust the hyperbolic component by setting different $\lambda$ values in Equation (7). The results of the ablation study are presented in Appendix D.2.

## 4.2 Inference and Link Prediction over Ontologies

**Benchmark** We utilize three normalized biomedical ontologies: GALEN Rector et al. (1996), Gene Ontology (GO) Ashburner et al. (2000), and Anatomy (Uberon) Mungall et al. (2012). As in Jackermeier et al. (2024), we use the entire ontology for training, and the complete set of inferred class subsumptions for testing. Those subsumptions can be regarded as partial order pairs $u \prec v$. Evaluation is performed using 1,000 subsumptions randomly sampled from the test set. Similar to inference over DAG, negative samples are generated by randomly replacing the child of each positive pair 10 times.

**Baselines** We focus on the most representative ontology embedding methods: ELBE Peng et al. (2022) and ELEM Kulmanov et al. (2019), as well as their enhanced versions incorporating RegD or $\tau$Box. Other hierarchy embedding methods are excluded from our tests due to their incompatibility

Table 3: F1 score (%) for the inference task.

| Method | GALEN | | GO | | ANATOMY | |
|---|---|---|---|---|---|---|
| | d=5 | d=10 | d=5 | d=10 | d=5 | d=10 |
| ELBE | 20.7 | 21.2 | 36.9 | 42.4 | 43.1 | 43.0 |
| + $\tau$Box | 20.8 | 20.7 | 32.2 | 34.7 | 42.2 | 47.2 |
| + RegD | **25.2** | **25.8** | **50.0** | **50.5** | **58.7** | **62.5** |
| ELEM | 16.9 | 17.3 | 23.5 | 27.4 | 34.6 | 38.7 |
| + RegD | **19.2** | **18.8** | **36.4** | **40.0** | **52.5** | **55.5** |

Table 4: F1 score (%) for the prediction task.

| Method | GALEN | | GO | | ANATOMY | |
|---|---|---|---|---|---|---|
| | d=5 | d=10 | d=5 | d=10 | d=5 | d=10 |
| ELBE | **21.0** | **24.2** | 32.8 | 37.9 | 25.1 | 25.5 |
| + $\tau$Box | 18.4 | 19.5 | 24.0 | 29.3 | 25.6 | 22.9 |
| + RegD | 20.6 | 21.0 | **37.1** | **44.1** | **41.4** | **45.3** |
| ELEM | 16.7 | 16.6 | 54.5 | 54.2 | **23.1** | **26.0** |
| + RegD | **16.8** | **18.0** | **60.3** | **61.4** | 21.7 | 21.9 |

with ontology embeddings. For example, OE and EntailmentCone utilizes cones as embedding objects, which cannot be directly integrated with ELBE or ELEM for ontology tasks. Details of the integration are provided in Appendix E.1.

**Results** The results are summarized in Tables 3. We can see that RegD yields consistent improvements across all ontologies for inference tasks. Specifically, it gains an F1 score increase of more than 45% with ELBE method on the ANATOMY ontology. Conversely, the $\tau$Box plugin consistently reduces performance across nearly all test cases, underscoring its limited applicability to tasks involving more than hierarchies.

### 4.3 LINK PREDICTION OVER ONTOLOGIES

We use the same baselines and datasets as described in Section 4.2. However, in this prediction task, we partition the original ontologies directly into 80% for training, 10% for validation, and 10% for testing as in Jackermeier et al. (2024). For the link prediction task, we focus on specific parts of the validation and testing sets, represented as $\exists r.B \prec ?A$, where $A$ and $B$ are concept names and $r$ is a role. This setup is equivalent to link prediction tasks $(?A, r, B)$ in knowledge graphs if we regard $A$, $B$, and $r$ as the head entity, tail entity, and relation, respectively.

**Results** Table 3 summarizes the results. RegD shows mixed results: while it generally improves performance, it decreases scores on the GALEN ontology and ANATOMY ontology with ELEM. This degradation likely occurs in challenging prediction cases where all methods perform poorly. Nevertheless, RegD achieves significant improvements in other cases, notably increasing F1 score by 77.6% with ELBE on the ANATOMY ontology. In contrast, the $\tau$Box plugin consistently reduces performance across almost all test cases.

### 5 CONCLUSION

We introduced a framework RegD for low-dimensional embeddings of hierarchies, leveraging two dissimilarity metrics between regions. Our method, applicable to regions in the Euclidean space, demonstrates versatility and has the potential for a wide range of tasks involving data beyond hierarchies. Additionally, we showed that our approach achieves comparable embedding performance to hyperbolic methods while being significantly simpler to implement.

For future work, we are interested in integrating our method with other approaches based on hyperbolic spaces. For instance, Hyperbolic Neural Networks Ganea et al. (2018a) currently depend on intricate manifold operations, such as parallel transport and maps between manifold and tangent spaces, to perform matrix multiplications and vector additions. In contrast, our framework enables these computations to be carried out more directly through the parameter space, which could eliminate the need for such complex manifold operations and thus simplify the implementation. Another promising direction for future work is to investigate replacing existing hyperbolic embedding components in related methods with our approach, including recent studies that couple large language models with hyperbolic embeddings to learn semantic hierarchies He et al. (2024b) , which can be extended from atomic concept to complex ones by integrating the ontology embeddings as in Section 4.2. Also, we are interested in generalizing our framework to support a wider variety of

region types—beyond the balls and boxes considered here—so that it can be better adapted to the requirements of diverse downstream tasks.

**Reproducibility statement** All code and data are publicly available at `https://anonymous.4open.science/r/RegD-F4E3`, along with clear instructions on environment configuration and hyperparameter settings to enable full reproduction of our results.

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

## A PROOFS

### A.1 THEOREM 1

*Proof of Theorem 1.* We begin by proving the result for balls and addressing each item as follows:

1. Let $reg_1 = ball(\mathbf{c}_1, r_1)$ and $reg_2 = ball(\mathbf{c}_2, r_2)$. For any ball $ball(\mathbf{c}', r') \subseteq ball(\mathbf{c}_2, r_2)$ with $r' \leq 0.5 \cdot r_1$, we must have:

$$||\mathbf{c}_1 - \mathbf{c}'||_p^p + |r_1 - r'|^p > (0.5 \cdot r_1)^p.$$

Therefore, we have:

$$d_{\text{dep}}(reg_1, ball(\mathbf{c}', r')) = \frac{||\mathbf{c}_1 - \mathbf{c}'||_p^p + |r_1 - r'|^p}{f(reg_1)f(ball(\mathbf{c}', r'))} > \frac{(0.5 \cdot r_1)^p}{f(reg_1)f(ball(\mathbf{c}', r'))}.$$

Thus, when:

$$f(ball(\mathbf{c}', r')) < \epsilon := \frac{(0.5 \cdot r_1)^p}{f(reg_1)[d_{\text{dep}}(reg_1, reg_2) + \Delta]},$$

we can guarantee that $d_{\text{dep}}(reg_1, ball(\mathbf{c}', r')) > d_{\text{dep}}(reg_1, reg_2) + \Delta$.

Note that by our assumption, we have:

$$\lim_{reg \to \emptyset} f(reg) = 0.$$

Therefore, there exists a $\delta > 0$ such that when $r < \delta$, we have $f(reg) \leq \epsilon$ for any region $reg = ball(\mathbf{c}, r)$. In conclusion, $r_0 = \min\{\delta, 0.5 \cdot r_1\}$ satisfies the required condition.

2. Let $reg = ball(\mathbf{c}, r)$. For any $n, M > 0$, we can select $n$ distinct vectors $\mathbf{c}_1, \ldots, \mathbf{c}_n \in ball(\mathbf{c}, 0.5 \cdot r)$ such that:

$$\|\mathbf{c}_i - \mathbf{c}_j\|_p^p > \delta > 0 \quad \text{for some} \quad \delta > 0.$$

Similarly to item 1, we can choose $r_i$ small enough such that for any ball $ball(\mathbf{c}_i, r_i)$, we have:

$$f(ball(\mathbf{c}_i, r_i)) < \left(\frac{\delta}{M}\right)^{0.5}.$$

Thus, we obtain:

$$d_{\text{dep}}(ball(\mathbf{c}_i, r_i), ball(\mathbf{c}_j, r_j)) > \frac{\delta}{f(ball(\mathbf{c}_i, r_i)) f(ball(\mathbf{c}_j, r_j))} \geq M.$$

This concludes the proof of item 2.

The proof for the case of boxes follows the same reasoning, with the radius $r$ replaced by the offset $\mathbf{o}$ or its norm $\|\mathbf{o}\|$. □

## A.2 PROPOSITION 1

*Proof of Proposition 1.* Recall that when the curvature is $-1$, the distance in the hyperbolic space (half-plane model) is given by:

$$d_k(\mathbf{x}, \mathbf{y}) = \text{arcosh}\left(1 + \frac{\|\mathbf{x} - \mathbf{y}\|_2^2}{2\mathbf{x}_n \mathbf{y}_n}\right).$$

Then, the distance induced by the function $F$ is of the form:

$$d_{\mathcal{B}^n}^{\#}(ball(\mathbf{c}, r), ball'(\mathbf{c}'.r')) = \text{arcosh}\left(1 + \frac{\|\mathbf{c} - \mathbf{c}'\|_2^2 + (r - r')^2}{2rr'}\right).$$

This coincides with the depth dissimilarity in Example 1 when $p = 2$, $g(x) = \text{arcosh}(x + 1)$, and $f(ball(\mathbf{c}, r)) = \sqrt{2}\, r$. This completes the proof. □

## A.3 PROPOSITION 2

*Proof of Proposition 2.* This proposition follows directly from Proposition 1. The function $h(x) = \text{arcosh}(x + 1)$ is an increasing bijection from $\mathbb{R}_{\geq 0}$ to $\mathbb{R}_{\geq 0}$. Thus, for any $x, x' \geq 0$, we have:

$$x \leq x' \iff h^{-1}(x) \leq h^{-1}(x').$$

By the assumption of this proposition, $d_{\text{dep}}(\cdot, \cdot) = k \cdot h^{-1}(d_{\mathbb{H}^n}(\cdot, \cdot))$, where $k > 0$. Therefore, for any points $\mathbf{x}_1, \mathbf{x}_2, \mathbf{x}_3, \mathbf{x}_4 \in \mathbb{H}^n$, we have:

$$d_{\mathbb{H}^n}(\mathbf{x}_1, \mathbf{x}_2) < d_{\mathbb{H}^n}(\mathbf{x}_3, \mathbf{x}_4)$$

if and only if

$$d_{\text{dep}}(F^{-1}(\mathbf{x}_1), F^{-1}(\mathbf{x}_2)) < d_{\text{dep}}(F^{-1}(\mathbf{x}_3), F^{-1}(\mathbf{x}_4)).$$

□

## A.4 PROPOSITION 3

*Proof of Proposition 3.* We prove each item one-by-one:

1. By definition, we have $d_{\text{bd}}(reg_1, reg_2) \leq 0$ if and only if $reg_1 \subseteq reg_2$. Next, we focus on the case $d_{\text{bd}}(reg_1, reg_2) = 0$.

   Note that if $d_{\text{bd}}(reg_1, reg_2) = 0$, we must have $reg_1 \subseteq reg_2$. Otherwise, we have

   $$d_{\text{bd}}(reg_1, reg_2) = \max_{\mathbf{x}_2 \in reg_2 \setminus reg_1} \{d(reg_1, \mathbf{x}_2)\} = 0$$

Therefore, for any $\mathbf{x}_2 \in reg_2$, we have $d(reg_1, \mathbf{x}_2) = 0$, therefore $\mathbf{x}_2 \subseteq reg_1$ (assuming $reg_1$ is a closed set). Contradiction!

Since $reg_1 \subseteq reg_2$, we have

$$d_{\text{bd}}(reg_1, reg_2) = \max_{\mathbf{x}_2 \in reg_2} \{-d(reg_1^c, \mathbf{x}_2)\} = 0.$$

Therefore, there must exist $\mathbf{x}_2 \in reg_2$ such that $d(reg_1^c, \mathbf{x}_2) = 0$, and thus $\mathbf{x}_2 \in reg_1^c$. Since we have $\mathbf{x}_2 \subseteq reg_2 \subseteq reg_1$, therefore, $\mathbf{x}_2 \in \partial(reg_1)$. Similarly, since $\mathbf{x}_2 \subseteq reg_1^c \subseteq reg_2^c$ and $\mathbf{x}_2 \in reg_2$, we also have $\mathbf{x}_2 \in \partial(reg_2)$. This finishes the proof of the first case.

2. By assumption, we have

$$d_{\text{bd}}(reg_1, reg_2) = \max_{\mathbf{x}_2 \in reg_2} \{-d(reg_1^c, \mathbf{x}_2)\}, \quad d_{\text{bd}}(reg_1, reg_2') = \max_{\mathbf{x}_2 \in reg_2'} \{-d(reg_1^c, \mathbf{x}_2)\}.$$

Since $reg_2' \subseteq reg_2$, of course we have

$$d_{\text{bd}}(reg_1, reg_2') \leq d_{\text{bd}}(reg_1, reg_2).$$

This finishes the proof of the second case.

$\square$

## A.5 GENERAL PARAMETERIZED REGIONS

A set of **parameterized regions** over Euclidean Space $\mathbb{R}^n$ can be defined as all regions $reg$ of the following form:

$$reg(\boldsymbol{\theta}) = \{\mathbf{x} \in \mathbb{R}^n : f_i(\mathbf{x}, \boldsymbol{\theta}) \leq 0, \quad \text{for } i = 1, \ldots, n\}, \tag{9}$$

where $f_1, \ldots, f_n$ are given (differentiable) functions from the given space to $\mathbb{R}$ and $\boldsymbol{\theta} = (\theta_1, \ldots, \theta_m) \in \mathbb{R}^m$ is the $m$-dimensional parameter. Let $\mathcal{R}$ denote a space of parameterized regions $reg(\boldsymbol{\theta})$ as in Equation 9, with $\boldsymbol{\theta} = (\theta_1, \ldots, \theta_m) \in \mathbb{R}^m$. For simplicity, and without significant loss of generality, in this section, we assume that all parameterized regions spaces $\mathcal{R}$ satisfies the following properties:

- **Volume-like parameter.** The last coordinate $\theta_m$ of $\boldsymbol{\theta}$ serves as a volume parameter such that $\theta_m > 0$ and $\lim_{reg(\boldsymbol{\theta}) \to \emptyset} \theta_m = 0$

- **Non-empty regions:** For any $\boldsymbol{\theta} \in \mathbb{R}^m$ with $\theta_m > 0$, the corresponding region $reg(\boldsymbol{\theta})$ is non-empty.

- **Uniqueness:** Two regions coincide if and only if their parameters are equal, i.e., $reg(\boldsymbol{\theta}) = reg(\boldsymbol{\theta}') \iff \boldsymbol{\theta} = \boldsymbol{\theta}'$.

- **Contractible:** For any parameter $\boldsymbol{\theta}$, there exists a sequence $\{\boldsymbol{\theta}^k = (\theta_1^k, \ldots, \theta_m^k)\}_{k=0}^{\infty}$ such that $\boldsymbol{\theta}^0 = \boldsymbol{\theta}$, $\theta_m^k > 0$ for all $k \geq 0$, $reg(\boldsymbol{\theta}^{k+1}) \subset reg(\boldsymbol{\theta}^k)$ for all $k \geq 0$, and $\lim_{k \to \infty} reg(\boldsymbol{\theta}^k) = \emptyset$.

It is worth noting that the spaces of balls and boxes in Examples 1 and 2 satisfy all of the above properties.

Notably, the results established in Theorem 1 naturally generalize to other parameterized regions, with proofs following analogous arguments. The result for the general case is of the following form.

**Theorem 2.** *Consider the parameterized region space $\mathcal{R}$ satisfies assumptions above, and equipped with the depth dissimilarity as defined in Definition 1. The following properties hold:*

*1. For any $reg_1, reg_2 \in \mathcal{R}$ and any $\Delta > 0$, there exists $reg_1$ sufficiently small, such that*

$$d_{dep}(reg_1, reg') > d_{dep}(reg_1, reg_2) + \Delta, \quad \forall reg' \subseteq reg_2.$$

*2. For any $reg \in \mathcal{R}$, any integer $n$, and any $M > 0$, there exist subsets $reg_1, \ldots, reg_n \subseteq reg$ such that for any distinct $i, j \in \{1, \ldots, n\}$,*

$$d_{dep}(reg_i, reg_j) > M.$$

*Outline of proof.* We sketch the main ideas, which closely follow the argument in the proof of Theorem 1.

For the first statement, by selecting a sufficiently small subset $reg' \subseteq reg_2$, we can ensure that

$$\|P(reg) - P(reg_1)\| > \delta$$

for some $\delta > 0$. Additionally, since $f(reg')$ can be made arbitrarily small as $reg' \to \emptyset$ (i.e., $\lim_{reg \to \emptyset} f(reg) = 0$), we can make the numerator of the depth dissimilarity arbitrarily small while keeping the denominator bounded below by a positive constant. Consequently, the depth dissimilarity $d_{\text{dep}}(reg_1, reg')$ can be made arbitrarily large.

For the second statement, a similar argument applies: by choosing the subsets $reg_1, \ldots, reg_n$ to be sufficiently small and mutually separated within $reg$, we can ensure that each $f(reg_i)$ is small enough so that the depth dissimilarity between any pair $(reg_i, reg_j)$ exceeds $M$.

□

Propositions 1 and 3 can also be extended to arbitrary regions as follows.

**Proposition 4.** *Let $\mathbb{H}^m$ denote the hyperbolic space with curvature $-1$. Assume that $\mathcal{R}$ is a collection of parameterized regions satisfying the above assumptions, equipped with the depth dissimilarity defined in Equation equation 1. Then the map $F \colon \mathcal{R} \to \mathbb{H}^m$ defined by $F(\boldsymbol{\theta}) = \boldsymbol{\theta}$ is a* bijective isometry *between $\mathcal{R}$ and $\mathbb{H}^m$ when $p = 2$, $g(x) = \text{arcosh}(x + 1)$, and $f(reg(\boldsymbol{\theta})) = \sqrt{2}\,\theta_m$.*

*Proof.* Under the given condition, the distance in the $\mathbb{H}^{n+1}$ is given by:

$$d_k(\mathbf{x}, \mathbf{y}) = \text{arcosh}\left(1 + \frac{\|\mathbf{x} - \mathbf{y}\|_2^2}{2\mathbf{x}_n \mathbf{y}_n}\right).$$

Then, the distance induced by the function $F$ is of the form:

$$d_{\mathcal{R}}^{\#}(reg(\boldsymbol{\theta}), reg(\boldsymbol{\theta}')) = \text{arcosh}\left(1 + \frac{\|\boldsymbol{\theta} - \boldsymbol{\theta}'\|_2^2}{2\theta_m \theta_m'}\right).$$

This coincides with the depth dissimilarity in Equation 1 when $p = 2$, $g(x) = \text{arcosh}(x + 1)$, and $f(reg(\boldsymbol{\theta})) = \sqrt{2}\,\theta_m$. □

**Proposition 5.** *Following Proposition 4, let the depth dissimilarity $d_{dep}$ be redefined by replacing $g$ to $g(x) = k \cdot x \, (k > 0)$. Then the map $F$ retains the following monotonicity property: for any points $\mathbf{x}_1, \mathbf{x}_2, \mathbf{x}_3, \mathbf{x}_4 \in \mathbb{H}^n$, $d_{\mathbb{H}^n}(\mathbf{x}_1, \mathbf{x}_2) < d_{\mathbb{H}^n}(\mathbf{x}_3, \mathbf{x}_4)$ if and only if*

$$d_{dep}(F^{-1}(\mathbf{x}_1), F^{-1}(\mathbf{x}_2)) < d_{dep}(F^{-1}(\mathbf{x}_3), F^{-1}(\mathbf{x}_4)).$$

*Proof.* The proof is the same as the proof of Proposition 3 in Appendix A.4 because of the function $h(x) = \text{arcosh}(x + 1)$ is an increasing bijection from $\mathbb{R}_{\geq 0}$ to $\mathbb{R}_{\geq 0}$. □

# B  HYPERBOLIC BOUNDARY DISSIMILARITY FOR BALLS

A ball in $\mathbb{R}^n$ can be mapped to a cone in the upper half-space $\mathbb{R}^n \times \mathbb{R}_{\geq 0}$ via a mapping $G$. This transformation is illustrated in Figure 4 for the case $n = 1$, where a ball $ball(0, 1)$ is mapped to a cone with apex $(0, 1) \in \mathbb{R} \times \mathbb{R}_{\geq 0}$. Note that the height of the cone corresponds to the radius of the underlying ball. The upper half-space can be interpreted as the Poincaré half-plane model of hyperbolic space.

Formally, we are considering a cone with a base as a ball $ball(\mathbf{c}, r)$ and a height $k > 0$, which can be defined as:

$$Cone(ball(\mathbf{c}, r), k) = \{(1 - t)\mathbf{x} + t(\mathbf{c} + k\mathbf{e}_{n+1}) : \mathbf{x} \in ball(\mathbf{c}, r), t \in [0, 1]\},$$

where $\mathbf{e}_{n+1} = (0, \ldots, 0, 1) \in \mathbb{R}^{n+1}$. The mapping $G$ is defined by:

$$G(ball(\mathbf{c}, r)) = Cone(ball(\mathbf{c}, r), r).$$

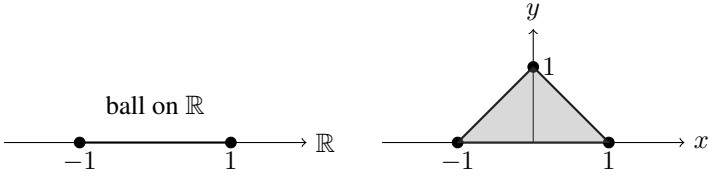

Figure 4: Mapping from balls to cones in the case of dimension 1.

The boundary dissimilarity between two cones in the Poincaré half-plane model (as shown on the right of Figure 4) can be defined as:

$$d_{\text{bd}}^{\text{cone}}(Cone_1, Cone_2) := \chi \cdot \min d_{\mathbb{H}}(\partial(Cone_1), \text{Apex}(Cone_1)),$$

where $\chi = -1$ if $Cone_2 \subseteq Cone_1$, and $\chi = 1$ otherwise. $\partial(Cone_1)$ denotes the boundary of $Cone_1$, and $\text{Apex}(Cone_1)$ is the apex of $Cone_1$, which is a single point.

Assuming the curvature is fixed at $-1$ (*i.e.*, $k = 1$), we specialize Theorem 4.2 from Yu et al. (2024) by imposing the condition $\sinh(\sqrt{k}\, r) = 1$, from which we derive:

$$d_{\text{bd}}^{\text{cone}}(Cone_1(ball(\mathbf{c}_1, r_1), r_1), Cone_2(ball(\mathbf{c}_2, r_2), r_2)) = \text{arcsinh}\left(\frac{\|\mathbf{c}_1 - \mathbf{c}_2\| - r_1}{r_2}\right) + \text{arcsinh}(1),$$

where $\mathbf{c}_i$ and $r_i$ are the center and radius of the underlying balls corresponding to the cones.

This distance can be extended back to the context of balls, allowing the definition of a boundary dissimilarity between balls as:

$$d_{\text{bd}}^{\text{cone}}(ball_1(\mathbf{c}_1, r_1), ball_2(\mathbf{c}_2, r_2)) = \text{arcsinh}\left(\frac{\|\mathbf{c}_1 - \mathbf{c}_2\| - r_1}{r_2}\right) + \text{arcsinh}(1).$$

## C  POINCARÉ BALL MODELS

There exist multiple models $\mathbb{H}$ that are isometric to each other. This work uses two such models, the Poincaré ball and the Poincaré half-space:

The **Poincaré ball** is given by

$$B^n = \{x \in \mathbb{R}^n : \|x\| < 1/\sqrt{k}\}.$$

Distances on $B^n$ are defined as

$$d_k(x, y) = \frac{1}{\sqrt{k}} \text{arcosh}\left(1 + \frac{2k\|x - y\|^2}{(1 - k\|x\|^2)(1 - k\|y\|^2)}\right).$$

## D  EXPERIMENTS SETTING

### D.1  HYPERPARAMETERS

Following the setup in Yu et al. (2024), we evaluated our models in dimensions $5$ and $10$. The weighting coefficient $\lambda$ in Eq. 7 was selected from $\{0, 0.5, 1\}$, while the learning rate was tuned over $\{0.01, 0.005, 0.001, 0.0005\}$. For the small dataset of mammals, which contains approximately 1000 nodes, we choose batch sizes from $\{32, 64, 128\}$. For the remaining datasets, we opt for batch sizes from $\{1024, 2048, 8192\}$. For the depth dissimilarity, we tested $p \in \{1, 2\}$ as specified in Definition 2. The boundary dissimilarity margins in Eq. 4 were set to $\gamma_1 = 0.001$ and $\gamma_2 = 0$ for directed acyclic graph (DAG) embedding tasks. In ontology embedding experiments, both margins were set to zero.

For DAG experiments, as in Yu et al. (2024), we trained for 400 epochs on the mammal and noun datasets and 500 epochs on the MCG and Hearst datasets. For ontology embedding experiments, we

Table 5: F1 score (%) on Mammal, WordNet noun, MCG, and Hearst with best numbers **bolded**.

| Options | | | | Mammal | | Noun | | MCG | | Hearst | |
|---|---|---|---|---|---|---|---|---|---|---|---|
| region | $\lambda$ | p | boundary dissimilarity | d=2 | d=5 | d=5 | d=10 | d=5 | d=10 | d=5 | d=10 |
| Box | 0 | - | $d_{\mathrm{bd}}$ | 39.4 | 42.8 | 34.6 | 34.6 | 49.9 | 56.6 | 41.6 | 44.7 |
| | | | $d_{vol}$ | 31.7 | 38.1 | 30.8 | 31.9 | 48.6 | **58.5** | **42.8** | **49.6** |
| | > 0 | 1 | $d_{\mathrm{bd}}$ | **64.9** | 71.6 | 53.8 | 51.3 | 43.7 | 45.0 | 35.6 | 38.8 |
| | | | $d_{vol}$ | 45.6 | 54.3 | 30.4 | 33.4 | 30.9 | 39.9 | 32.6 | 38.8 |
| | | 2 | $d_{\mathrm{bd}}$ | 56.1 | 62.2 | 50.3 | 47.4 | **50.7** | 57.4 | 40.6 | 44.6 |
| | | | $d_{vol}$ | 39.7 | 47.9 | 30.6 | 31.6 | 48.5 | 56.7 | 37.8 | 43.8 |
| Ball | 0 | - | $d_{\mathrm{bd}}$ | 42.8 | 55.5 | 39.6 | 42.5 | 43.9 | 46.7 | 37.7 | 37.1 |
| | | | $d_{\mathrm{bd}}^{\mathrm{cone}}$ | 41.2 | 49.7 | 36.4 | 37.7 | 40.9 | 43.7 | 36.0 | 36.7 |
| | > 0 | 1 | $d_{\mathrm{bd}}$ | 58.2 | 69.2 | 51.7 | 54.4 | 40.9 | 45.9 | 33.6 | 35.8 |
| | | | $d_{\mathrm{bd}}^{\mathrm{cone}}$ | 57.9 | 65.3 | 52.3 | 54.7 | 36.8 | 41.7 | 31.6 | 34.6 |
| | | 2 | $d_{\mathrm{bd}}$ | 62.5 | **71.8** | **58.4** | **59.1** | 41.7 | 45.8 | 34.8 | 37.7 |
| | | | $d_{\mathrm{bd}}^{\mathrm{cone}}$ | 62.7 | 67.7 | 54.3 | 55.1 | 40.7 | 45.7 | 34.6 | 36.7 |

trained for 5000 epochs, following the protocol in the previous work Jackermeier et al. (2024) of ontology embedding.

All experiments were conducted on a system equipped with an AMD Ryzen Threadripper PRO 7965WX 24-core processor, 128GB of RAM, and an NVIDIA A6000 GPU with 48GB of VRAM, running Ubuntu 24.04.

## D.2 DETAILED RESULTS

The detail results under different hyperparameter choices of $\lambda$, $p$ and boundary dissimilarities are summarized in Table 5. Note that, when $\lambda = 0$, the depth dissimilarity component is absent from the energy function, and therefore, the choice of $p$, as a hyperparameter of depth dissimilarity, is not applicable.

From the results, it is evident that our boundary dissimilarity generally outperforms the specific boundary dissimilaritys for boxes and balls (i.e., $d_{vol}$ and $d_{\mathrm{bd}}^{\mathrm{cone}}$) in most cases. Furthermore, when using boxes as the embedding object, setting $p = 1$ yields better results than $p = 2$. However, for balls, $p = 2$ achieves superior performance.

For the mammal and noun datasets, using balls as the embedding object and incorporating the depth dissimilarity (i.e., $\lambda > 0$) in the energy function results in better performance. Conversely, for the MCG and Hearst datasets, the best performance is typically achieved when using boxes as the embedding region and setting $\lambda = 0$, effectively excluding the hyperbolic components. This observation aligns with our analysis in the main text.

It is worth noting, as shown in the seventh row of Table 5, that even under a small configuration range (i.e., with box and $\lambda \in 0.5, 1$), our method outperforms existing approaches and exhibits only minimal performance gaps compared to the best results obtained with other configurations.

## D.3 OTHER RESULTS

**Including non-basic edges in training** We evaluated the impact of including different percentages of transitive closure edges in the training set using the mammal dataset. The results, presented in Table 6, demonstrate that our method achieved overall superior performance compared to existing approaches. This was particularly evident when the dimension was 2, where our method outperformed all other approaches across all conditions except with 25% non-basic edges—and even then, our F1-score was only 0.2 lower than the best performer. Our method exhibited more stable performance characteristics, with F1-scores increasing gradually as we added non-basic edges to the training set. In contrast, the current state-of-the-art method, ShadowCone, showed sharp initial

Table 6: F1 score (%) on mammal sub-graph with best numbers **bolded**. Results with * are coming from Yu et al. (2024). For ShadowCone, only result of Umbral-half-space is reported as in Yu et al. (2024).

| Non-basic-edge | Dimension = 2 | | | | | Dimension = 5 | | | | |
|---|---|---|---|---|---|---|---|---|---|---|
| Percentage | 0% | 10% | 25% | 50% | 90% | 0% | 10% | 25% | 50% | 90% |
| GBC-box* | 23.4 | 25.0 | 23.7 | 43.1 | 48.2 | 35.8 | 60.1 | 66.8 | 83.8 | **97.6** |
| VBC-box* | 20.1 | 26.1 | 31.0 | 33.3 | 34.7 | 30.9 | 43.1 | 58.6 | 74.9 | 69.3 |
| $\tau$Box | 29.0 | 33.0 | 41.2 | 49.6 | 53.5 | 33.5 | 37.5 | 45.0 | 58.9 | 64.0 |
| EntailmentCone* | 54.4 | 61.0 | 71.0 | 66.5 | 73.1 | 56.3 | 81.0 | 84.1 | 83.6 | 82.9 |
| ShadowCone * | 57.7 | 73.7 | **77.4** | 80.3 | 79.0 | 69.4 | **81.1** | 83.7 | **88.5** | 91.8 |
| Ours (box) | **64.9** | **74.4** | 75.8 | 78.3 | 82.8 | 71.6 | 77.8 | 83.2 | 87.6 | 87.0 |
| Ours (ball) | 62.7 | 68.3 | 77.2 | **84.1** | **88.6** | **71.8** | 74.5 | **84.4** | 88.4 | 90.2 |

Table 7: Average and standard deviation (Std) of 10 random runs of F1 score (%) on Mammal, WordNet noun, MCG, and Hearst.

| RegD | | Mammal | | Noun | | MCG | | Hearst | |
|---|---|---|---|---|---|---|---|---|---|
| | | d=2 | d=5 | d=5 | d=10 | d=5 | d=10 | d=5 | d=10 |
| fixed seed | box | **64.9** | 71.6 | 53.8 | 51.3 | **50.7** | **58.5** | **42.8** | **49.6** |
| | ball | 62.7 | **71.8** | **58.4** | **59.1** | 44.9 | 46.8 | 37.7 | 37.7 |
| average ($\pm$std) | box | **63.00** ($\pm$2.01) | **69.65** ($\pm$3.14) | **52.98** ($\pm$1.00) | 49.33 ($\pm$1.12) | **50.49** ($\pm$0.44) | **57.71** ($\pm$0.41) | **42.70** ($\pm$0.50) | **48.72** ($\pm$0.12) |
| average ($\pm$std) | ball | 56.70 ($\pm$4.91) | 66.76 ($\pm$2.91) | 51.63 ($\pm$1.52) | **59.31** ($\pm$2.29) | 44.09 ($\pm$0.55) | 46.35 ($\pm$0.37) | 37.24 ($\pm$0.37) | 37.11 ($\pm$0.47) |

improvements followed by diminishing returns. Specifically, when increasing non-basic edges from 0% to 10%, ShadowCone's F1-scores jumped dramatically (increasing by 16.0 and 11.7 points for dimensions 2 and 5, respectively). However, further increases from 10% to 25% yielded much smaller improvements (3.7 and 2.6 points for dimensions 2 and 5, respectively).

**Repeatness** The main experiments presented in this paper were conducted using a fixed random seed, following established practices in previous works Yu et al. (2024); Ganea et al. (2018b). To assess the repeatability and robustness of our results, we conducted 10 independent experimental runs on each of the four datasets. The average performance and standard deviation are reported in Table 7, while the outcomes of individual runs are summarized in Figure 5. These results demonstrate that our method exhibits considerable stability, particularly on the MCG and Hearst datasets. Compared to the main results obtained with a fixed random seed, the overall average performance remains very close, with only a few exceptions:

1. In certain cases—such as using balls on the Mammal dataset with $d = 2$ and on the Noun dataset with $d = 5$—performance is slightly lower, though still superior to that of other baseline methods.

2. In some settings, performance is even improved on average, for example, when using balls on the Noun dataset with $d = 10$.

These results indicate that the proposed method is robust across different random seeds, with consistent performance across most settings.

**Impact of $\lambda$** In Table 5, we evaluate the performance of $\lambda = 0$, 0.5, and 1, and report results for both $\lambda = 0$ and $\lambda > 0$. To provide a more detailed analysis, Table 8 presents additional experiments with $\lambda$ ranging from 0 to 1 in increments of 0.1, using the mammal dataset (dim = 2, box embedding).

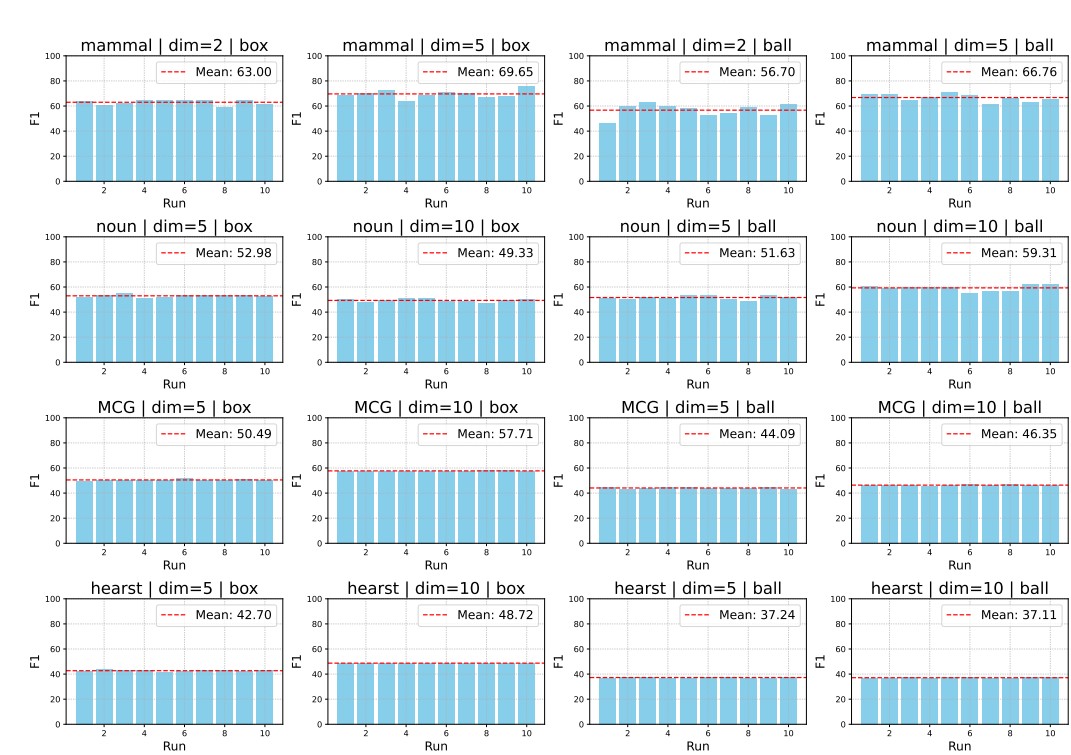

Figure 5: All F1 scores of 10 random runs on Mammal, WordNet noun, MCG, and Hearst.

The results suggest that the impact of $\lambda$ may not follow a clear or consistent trend, and the optimal value of $\lambda$ can vary significantly across different settings. For instance, in this dataset, larger values of $\lambda$ generally lead to higher F1 scores compared to smaller ones. However, the relationship is not strictly monotonic—an increase in $\lambda$ does not always result in improved performance.

| | 0 | 0.1 | 0.2 | 0.3 | 0.4 | 0.5 | 0.6 | 0.7 | 0.8 | 0.9 | 1 |
|---|---|---|---|---|---|---|---|---|---|---|---|
| **F1**(avg) | 42.70 | 50.92 | 57.42 | 56.07 | 57.03 | 64.43 | 62.46 | 62.83 | 66.56 | 62.80 | 65.30 |
| (±std) | (±7.07) | (±5.44) | (±5.46) | (±4.87) | (±5.62) | (±4.61) | (±3.23) | (±2.11) | (±2.33) | (±2.47) | (±2.08) |

Table 8: F1 scores with standard deviations for different $\lambda$ values on the mammal dataset (dim=2, using box).

# E   ONTOLOGIES

Ontologies use sets of statements (axioms) about concepts (unary predicates) and roles (binary predicates) for knowledge representation and reasoning. We focus on $\mathcal{EL}$-ontologies, which strike a good balance between expressivity and reasoning efficiency, making them widely applicable Baader & Gil (2024).

Let $\mathsf{N_C} = \{A, B, \dots\}$, $\mathsf{N_R} = \{r, t, \dots\}$, and $\mathsf{N_I} = \{a, b, \dots\}$ be pairwise disjoint sets of *concept names* (also called *atomic concepts*), *role names*, and *individual names*, respectively. $\mathcal{EL}$-*concepts* are recursively defined from atomic concepts, roles, and individuals as follows:

$$\top \mid \bot \mid A \mid C \sqcap D \mid \exists r.C \mid \{a\}$$

An $\mathcal{EL}$-*ontology* is a finite set of TBox axioms of the form

$$C \sqsubseteq D.$$

Note that here $\sqsubseteq$ denotes "SubclassOf", which is a transitive relation that can be considered a partial order $\prec$ by rewriting it as:

$$C \sqsubseteq D \quad \Leftrightarrow \quad D \prec C.$$

**Example 4.** *From atomic concepts Father, Child, Male, . . . and the role hasParent, we can construct a small family ontology consisting of two TBox axioms:*

$$Father \sqsubseteq Male \sqcap Parent, \quad Child \sqsubseteq \exists hasParent.Father.$$

An *interpretation* $\mathcal{I} = (\Delta^{\mathcal{I}}, \cdot^{\mathcal{I}})$ consists of a non-empty set $\Delta^{\mathcal{I}}$ and a function $\cdot^{\mathcal{I}}$ that maps each $A \in \mathsf{N_C}$ to $A^{\mathcal{I}} \subseteq \Delta^{\mathcal{I}}$, and each $r \in \mathsf{N_R}$ to $r^{\mathcal{I}} \subseteq \Delta^{\mathcal{I}} \times \Delta^{\mathcal{I}}$, where $\bot^{\mathcal{I}} = \emptyset$ and $\top^{\mathcal{I}} = \Delta^{\mathcal{I}}$. The function $\cdot^{\mathcal{I}}$ is extended to any $\mathcal{EL}$-concepts as follows:

$$(C \sqcap D)^{\mathcal{I}} = C^{\mathcal{I}} \cap D^{\mathcal{I}}, \tag{10}$$

$$(\exists r.C)^{\mathcal{I}} = \{a \in \Delta^{\mathcal{I}} \mid \exists b \in C^{\mathcal{I}} : (a, b) \in r^{\mathcal{I}}\}. \tag{11}$$

An interpretation $\mathcal{I}$ *satisfies* a TBox axiom $X \sqsubseteq Y$ if $X^{\mathcal{I}} \subseteq Y^{\mathcal{I}}$ for $X$ and $Y$ being either two concepts or two role names, or $X$ being a role chain and $Y$ being a role name. An ontology $\mathcal{O}$ *entails* an axiom $C \sqsubseteq D$, written

$$\mathcal{O} \models C \sqsubseteq D \quad (\text{i.e., } \mathcal{O} \models D \prec C),$$

if $C \sqsubseteq D$ is satisfied by all models of $\mathcal{O}$. Ontologies can infer much more complex patterns than the transitive closure, as shown by the following example.

**Example 5.** *In an ontology, suppose that in a group $X$, all members are both a man and a parent. That is, $X \sqsubseteq$ man and $X \sqsubseteq$ parent in the ontology. Moreover, man and parent together imply father (formally, man $\sqcap$ parent $\sqsubseteq$ father in the ontology). From this, we can infer that the group $X$ is a father, which is not derivable from the transitive closure.*

### E.1 ONTOLOGY EMBEDDING METHODS

**Implementation of ELBE and ELEM over DAG**   As discussed in the main text, the ontology embedding methods ELBE and ELEM can be adapted to serve as baseline methods for embedding DAGs using boxes and balls, respectively. Specifically, for a pair of nodes $(u, v)$, the energy function used during training is defined as follows:

1. **ELBE:** Nodes $u$ and $v$ are embedded as boxes $box_u$ and $box_v$, respectively. The energy function is given by:

$$E(u, v) = \|\max\{|\mathbf{c}_u - \mathbf{c}_v| + \mathbf{o}_v - \mathbf{o}_u, \mathbf{0}\}\|,$$

where $\mathbf{c}_u$ and $\mathbf{c}_v$ denote the center vectors of the boxes, and $\mathbf{o}_u$ and $\mathbf{o}_v$ denote the offsets of the boxes.

2. **ELEM:** Nodes $u$ and $v$ are embedded as balls $ball_u$ and $ball_v$, respectively. The energy function is defined as:

$$E(u, v) = \max\{\|\mathbf{c}_u - \mathbf{c}_v\| + r_v - r_u, \mathbf{0}\},$$

where $\mathbf{c}_u$ and $\mathbf{c}_v$ represent the centers of the balls, and $r_u$ and $r_v$ represent their radii.

Note that the original ELEM method includes a regularization term that enforces the norms $\|\mathbf{c}_u\|$ and $\|\mathbf{c}_v\|$ to be close to 1. However, for the DAG case, we omit this regularization primarily applies to scenarios involving relation embeddings, such as axioms of the form $A \sqsubseteq \exists r.B$ or $\exists r.B \sqsubseteq A$, as in the KGE methods TransE Bordes et al. (2013).

**Integration of ELBE and ELEM with RegD and $\tau$Box**   ELBE and ELEM are embedding approaches for normalized $\mathcal{EL}$-ontologies, which consist of four types of axioms $C \sqsubseteq D$:

$$A \sqsubseteq B, \quad A \sqcap B \sqsubseteq B', \quad A \sqsubseteq \exists r.B, \quad \exists r.B \sqsubseteq A.$$

ELBE embeds each atomic concept $A$ as a box $box(A) \subseteq \mathbb{R}^n$ and maps complex concepts like $\exists r.B$ to a new box obtained by translating $box(B)$ by a vector $\mathbf{v}_r \in \mathbb{R}^n$:

$$box(\exists r.B) = \{\mathbf{x} - \mathbf{v}_r \mid \mathbf{x} \in box(B)\}$$

ELEM employs a similar approach but uses balls instead of boxes. The key differences between ELBE and ELEM are as follows:

1. ELEM includes a regularization term, similar to TransE, to constrain the norm of the center of each ball to be close to 1.

2. ELBE handles conjunctions more effectively, as the intersection of two boxes is still a box. This allows it to define $box(A \sqcap B)$ as $box(A) \cap box(B)$. In contrast, ELEM, which uses balls, cannot handle intersections directly and must use specialized mechanisms to approximate satisfaction of the axiom $A \sqcap B \sqsubseteq B'$.

To integrate RegD and $\tau$Box into ELBE or ELEM, it suffices to replace their energy function for axioms $C \sqsubseteq D$ with the corresponding energy function from RegD or $\tau$Box. However, ELBE and ELEM handle negative samples differently, using a loss function of the form:

$$\mathcal{L} = \sum_{(C,D)\in P} E(D,C) + \sum_{(C',D')\in N} (\gamma - E(D', C')).$$

When integrating with RegD, this loss function should also be replaced with the one defined in Section 3.3.

**Result of prediction task by ranking-based metrics**  The evaluation results for the prediction task on ontologies are presented in Table 9. RegD shows consistent improvements for both ELEM and ELBE across most cases. The gains are particularly notable for the GO and ANATOMY ontologies, where the H@100 score increases from 0 to 0.26. In contrast, performance on the GALEN ontology remains low for both variants (with or without RegD), likely due to the inherent complexity of GALEN. This suggests that more advanced methods may be required to effectively model such complex structures.

# F    THE USE OF LLMS

LLMs have been employed solely to assist with polishing the writing. All content was originally written by the authors and subsequently refined by LLMs to improve readability, clarity, and to correct grammatical errors or typographical mistakes.

Table 9: Prediction task

| Dataset | Method | H@1 | H@10 | H@100 | Medain | MRR | MR | AUC |
|---------|--------|-----|------|-------|--------|-----|-----|-----|
| GALEN | ELBE | 0 | 0 | 0 | 7831 | 0 | 9044 | 0.61 |
| (d=5) | + RegD | 0 | 0 | 0.02 | 9506 | 0 | 10056 | 0.57 |
| | ELEM | 0 | 0.01 | 0.04 | 8843 | 0 | 9361 | 0.60 |
| | + RegD | 0 | 0 | 0.02 | 10122 | 0 | 10531 | 0.55 |
| GALEN | ELBE | 0 | 0 | 0 | 8812 | 0 | 9595 | 0.59 |
| (d=10) | + RegD | 0 | 0.01 | 0.02 | 8966 | 0 | 9729 | 0.58 |
| | ELEM | 0 | 0 | 0.03 | 8856 | 0 | 9534 | 0.59 |
| | + RegD | 0 | 0.01 | 0.03 | 9996 | 0 | 10441 | 0.55 |
| GO | ELBE | 0 | 0 | 0 | 14058 | 0 | 17231 | 0.62 |
| (d=5) | + RegD | 0 | **0.10** | **0.19** | 8466 | 0.02 | 14376 | 0.69 |
| | ELEM | 0 | 0 | 0.01 | 19666 | 0 | 19813 | 0.57 |
| | + RegD | 0 | 0.01 | 0.03 | 15440 | 0 | 17983 | 0.61 |
| GO | ELBE | 0 | 0 | 0 | 18926 | 0 | 19656 | 0.57 |
| (d=10) | + RegD | 0 | **0.17** | **0.26** | 7785 | 0.04 | 14244 | 0.69 |
| | ELEM | 0 | 0 | 0.01 | 19668 | 0 | 20543 | 0.55 |
| | + RegD | 0 | 0.02 | 0.06 | 17194 | 0.01 | 19019 | 0.59 |
| ANATOMY | ELBE | 0 | 0 | 0.01 | 19556 | 0 | 29528 | 0.72 |
| (d=5) | + RegD | 0 | 0.02 | 0.05 | 14317 | 0.01 | 25582 | 0.76 |
| | ELEM | 0 | 0.01 | 0.04 | 23437 | 0 | 32788 | 0.69 |
| | + RegD | 0 | 0.04 | 0.14 | 8616 | 0.01 | 27272 | 0.74 |
| ANATOMY | ELBE | 0 | 0 | 0.03 | 14955 | 0 | 25844 | 0.76 |
| (d=10) | + RegD | 0 | **0.05** | **0.13** | 10019 | 0.02 | 22215 | 0.79 |
| | ELEM | 0 | 0.01 | 0.06 | 14573 | 0 | 27010 | 0.75 |
| | + RegD | 0 | **0.07** | **0.22** | 8066 | 0.03 | 27632 | 0.74 |

