# OpenReview forum: "Achieving Hyperbolic-Like Expressiveness with Arbitrary Euclidean Regions: A New Approach to Hierarchical Embeddings"
_ICLR.cc/2026/Conference — Submitted to ICLR 2026_

### Official Review · Reviewer_Lt9K · 2025-10-24

**Soundness:** 3
**Presentation:** 3
**Contribution:** 3
**Rating:** 6
**Confidence:** 4

**Summary:**

This paper introduces a new way of embedding hierarchical data, which represents instances as regions in Euclidean space instead of single points. The authors show that a special case of their framework results in a model that is isometric to hyperbolic space, thereby inheriting its beneficial properties. Through several experiments the authors show the strength of their method for embedding DAGs and ontologies compared to existing methods.

**Strengths:**

- The newly proposed method is in my opinion a very interesting approach that could potentially solve some of the issues that are usually encountered when dealing with hyperbolic space, while still retaining its benefits.
- The new model of hyperbolic space, implicated by Proposition 1, is a very interesting new perspective on hyperbolic geometry that could lead to many new insights.
- The theory supporting the method is convincing, well-structured and the proofs seem correct to me.
- Based on the results, the method seems to perform quite well on hierarchical data embedding.

**Weaknesses:**

My primary concern with this paper is that it makes a few strong claims that are not motivated well enough in my opinion:
- The claim is that the method can be applied with any kind of region as defined in Appendix A5. However, the theory all uses specific cases involving balls and boxes. To make a claim about arbitrary regions, the theory and proofs should be generalized to arbitrary regions.
- It is claimed that "only the relative order of the scores matters" based on the F1 score and Hits@k relying solely on the ranking. However, in other hierarchy embedding papers (e.g., [1, 2]) distance-based metrics such as the distortion are often reported as well. I think this should either be included in the evaluation or its exclusion should be motivated.
- The authors claim that their method and this framework are simpler then existing hyperbolic frameworks (lines 165-170 & lines 480-481). It is not completely clear to me why this is the case and I think it requires some more motivation.

I also noticed that the results in Table 2 seem to be a combination of the results from Table 5, picking the best configuration for most experiments. This seems a bit dishonest to me. If you want to pick the best configuration depending on the experiment, then these values should be considered hyperparameters and included in the hyperparameter search instead of being presented as an ablation.

Lastly, I think the paper could benefit from a bit more proofreading. For example, $g(x) = k \cdot x + b$ should probably be $g(x) = k \cdot x$ in Proposition 2, given Definition 1.

Despite these weaknesses I want to reiterate that I think the proposed method and the new perspective on hyperbolic geometry are interesting enough for the paper to be accepted. Moreover, I am willing to raise my score if my concerns are addressed.

[1] Sala, Frederic, et al. "Representation tradeoffs for hyperbolic embeddings." International conference on machine learning, 2018.
[2] van Spengler, Max, and Pascal Mettes. "Low-distortion and GPU-compatible Tree Embeddings in Hyperbolic Space." International Conference on Machine Learning, 2025.

**Questions:**

1. I'm a bit confused by the part in lines 329-344. These specialized metrics do not satisfy Proposition 3, right? So why would we want to use these?
2. Can you elaborate on the statement in line 480-481 regarding the simplification of the implementation of HNN a bit? I don't really see how the required operations would simplify in your model of hyperbolic space.

---

> ### Author Response · Authors · 2025-11-19
>
> Thank you for your review and feedback. We respond to your comments and questions below.
>
> ---
>
> **Weakness 1: Missing theoretical results for arbitrary regions**
>
> **Response:**
>
> The version of **Theorem 1** for arbitrary regions has been provided in Lines 785–805 (Appendix A5) in the new manuscript. We have also added a formal extension of **Propositions 1 and 2** to parameterized regions in Lines 810–845. Note that Proposition 2 was already stated and proved for arbitrary regions. With these additions, all theoretical results now include their corresponding extensions to the arbitrary-regions setting.
>
> ---
>
> **Weakness 2: Missing distance-based metrics**
>
> **Response:**
> Our evaluation follows EntailmentCones and ShadowCone, which focus on the inferred edges generated by the transitivity of the hierarchy and thus use metrics like F1-score. In contrast, the works [1,2] that use distortion focus on the reconstruction of the trees, instead of the inferred edges by the hierarchy, and are not directly comparable in our setting.
>
> ---
>
> **Weakness 3: Vague explanation of simplicity**
>
> **Response:**
> In Lines 165–170, "facilitates implementation" means there is no need for specific SGD on Riemannian manifolds and no requirement for using double-precision vectors as in the hyperbolic embedding case. "Facilitates potential extensions" means easy integration with other works, such as ontology embeddings. In Lines 480–481, it mainly means there is no need for manifold operations in Hyperbolic Neural Networks (detailed  in the answer to Question 2). We have added the explanation above in the new version of paper for better clarity.
>
> ---
>
> **Weakness 4: Inappropriate choice of results from ablation studies**
>
> **Response:**
> The contents of Table 5 are in fact results under different hyperparameters:
> - The columns for $p$ and $\lambda$ are hyperparameters, as stated in Lines 856 and 859 of Appendix D, respectively.
> - The column for boundary dissimilarity is also considered a hyperparameter, since we accommodate specialized boundary dissimilarities as described in Line 332.
>
> We acknowledge that the term *“Ablation Studies”* is inappropriate and have renamed it to *“Detailed Results”*.
>
> ---
>
> **Weakness 5: Typos and mistakes**
>
> **Response:**
> Thank you for pointing out the typos in Proposition 2. We are thoroughly proofreading the paper to eliminate remaining errors. For example, we have corrected typos such as “inferencing” and “it gains--” in Line 449, improved the sentence structure in Lines 785–790, and clarified the notation in Lines 736–744. All changes have been highlighted in blue.
>
> ---
>
> **Question 1: Inclusion of specific boundary dissimilarities**
>
> **Response:**
> The two specific constructions in Lines 329–344 actually satisfy Proposition 3, except for the conclusion that the dissimilarity equals 0 when one region is internally tangent to another for the volume-based dissimilarity in Equation (5). Therefore, we include them as a special case alongside our general construction in Lines 262–327 to make our framework more complete.
>
> ---
>
> **Question 2: Explaining how to simplify Hyperbolic Neural Networks**
>
> **Response:**
> Hyperbolic Neural Networks (HNNs) are built upon hyperbolic spaces, where standard neural network operations are not directly well-defined—for example, addition or matrix multiplication of points in hyperbolic space is not defined. To address this issue, HNNs rely on manifold operations such as the exponential and logarithmic maps ($ \exp_x(v), \log_x(y) $), which transfer points between the hyperbolic space and its tangent space (the latter being Euclidean).
>
> Since our framework is constructed within Euclidean space, we are interested in exploring whether it is possible to design an alternative version of HNNs that avoids such complex manifold operations, thereby simplifying the model. One possible idea is to apply vector addition or matrix multiplication directly to the region parameters, while capturing hierarchical properties through specially designed losses in the final layer or by pre-training the initial embeddings using depth and boundary dissimilarities.

---

> > ### Comment · Reviewer_Lt9K · 2025-11-25
> >
> > Thank you for your responses and for adding additional theory to the paper. Most of my concerns have been addressed, but there are still two points that I am unsure about:
> > 1. I think that some of the theory requires some stricter definitions of what a parameterized region is. For example, in the proof of Theorem 2 it is assumed that for any region you can pick an arbitrarily small region contained within it. However, this seems false to me. As an example, suppose we have a single function defining our set of parameterized regions: $f\_1(\mathbf{x}, r) = ||\mathbf{x}|| - (1 + r)$. Obviously this is not a particularly interesting set of regions, but it does seem to satisfy your definition of such a set. This example seems to contradict Theorem 2. The framework using arbitrary parameterized regions seems interesting to me, but since it is proposed as a formal mathematical framework, it should be completely consistent in my opinion.
> > 2. I don't fully understand the proposed motivation for the claim regarding the simplicity of neural networks within this framework. The authors claim that manifold operations can be avoided for example through the use of vector additions and matrix multiplications. However, I don't see what geometric interpretation such operations have within this framework and particularly not how these can be used to recreate a simplified hyperbolic neural network alternative. Is it possible to expand on this a bit to provide a more convincing sketch of how something in this direction could be achieved? If not, then I think it might be better to weaken the corresponding claims in the paper.

---

> ### Author Response · Authors · 2025-11-25
>
> Thanks for your reply! We would like to further clarify the points you raised:
>
> ---
>
> 1. Thank you for pointing this out. We agree that the property *“for any region one can pick an arbitrarily small region contained within it”*, while natural, does not necessarily hold for all parameter region spaces. We have revised the manuscript accordingly by explicitly adding this assumption, which we refer to as *"contractible"*, and moving all assumptions of the region space to the beginning of Appendix A.5 (previously in Lines 810–822).
>
> ---
>
> 2. For the geometric interpretation, we use balls as an illustrative example:
>    - Adding a vector to the center parameters translates the balls.
>    - Adding values to the radii uniformly expands or contracts the balls.
>    - Applying matrix transformations to the centers or radii induces linear transformations of the corresponding geometric attributes.
>
>     The overall workflow could be as follows: starting with randomly initialized embeddings in the parameterized-region space,   one can directly adopt standard multilayer neural networks (e.g., Transformer, RNN, etc) while treating different parameter components (e.g., centers or radii) differently if desired. Finally, task-specific constructions or loss functions can be incorporated to reflect the structure of the underlying region space, analogous to the HNN setting.
>
>     For example, in classification tasks, HNN replaces the standard softmax with probabilities defined via distances to hyperplanes in hyperbolic space. In a similar spirit, one could design losses that exploit the geometry of the parameterized-region space, potentially leveraging our depth- or boundary-based dissimilarities.
>
>     This provides a conceptually straightforward approach to integrating our framework with HNN. While it may not yet be immediately implementable, we believe it nevertheless points toward a promising direction for future research.

---

### Official Review · Reviewer_ki2e · 2025-10-28

**Soundness:** 3
**Presentation:** 3
**Contribution:** 3
**Rating:** 6
**Confidence:** 3

**Summary:**

The paper introduces RegD, a region-based Euclidean embedding framework that aims to match the expressiveness of hyperbolic methods for hierarchical data while remaining flexible enough to support arbitrary geometric regions. It defines two complementary dissimilarities: a depth dissimilarity  that scales pairwise separation by region size to emulate hyperbolic exponential growth, and a boundary dissimilarity that captures asymmetric inclusion  via distances to region boundaries.

**Strengths:**

The paper offers a clean, general framework that reproduces key benefits of hyperbolic embeddings while staying in Euclidean space, combining a size-aware depth dissimilarity with an asymmetric boundary dissimilarity to model both separation across levels and set-inclusion along hierarchies. The theory is aligned with practice (mostly), depth dissimilarity can emulate hyperbolic behavior and even preserve the ranking of hyperbolic distances with simple choices of g, which directly supports ranking-based metrics used in evaluation.

**Weaknesses:**

Proposition 1 is elegant but narrow, and the follow-up argument that only ranking matters sidesteps scale/geometry mismatches; there is no empirical study showing that different g choices, norms, or region parameterizations leave training dynamics and downstream behavior unchanged. Also boundary dissimilarity definition mixes containment and non-containment with different signs and extrema; for balls the presented closed form appears to omit an explicit case split, which can be confusing and invites edge-case ambiguities that are not stresstested experimentally.

**Questions:**

It is possible to use classical isometries between hyperbolic spaces to prove your results in a model agnostic context?

---

> ### Author Response · Authors · 2025-11-19
>
> Thank you for your review and feedback. We respond to your comments and questions below.
>
> ---
>
> **Weakness 1:  Narrow Proposition 1, ignore scale/geometry mismatches in  Proposition 2**
>
> **Response:**
>
> We have extended Propositions 1 and 2 to cover generalized, parameterized regions (in blue at Lines 810–845, Appendix A5) for addressing the concern of narrow applicability.
>
> Regarding scale/geometry mismatches:
> Proposition 2 is not intended to exactly replicate hyperbolic distances (as Proposition 1 does). Instead, it prioritizes the property most relevant to embedding quality—ranking consistency—and does not require exact scale or geometric alignment.
>
>
> ---
>
> **Weakness 2: Empirical study for different $ g $, norms, or region parameterizations**
>
> **Response:**
> A detailed result under different norms and region parameterizations is provided in Table 5, Appendix D. The results suggest that different norms or region parameterizations do influence downstream behavior, and the optimal choice may vary across datasets. Regarding $ g $, we focus on the simplest option—the identity function—which is sufficient for preserving the main properties of hyperbolic spaces, as discussed in our response to Weakness 1.
>
>
>
> ---
>
> **Weakness 3: Mixing containment and non-containment in boundary dissimilarity**
>
> **Response:**
> We chose this setting to better distinguish containment and non-containment directly through the values. Of course, it is also possible to represent all values as positive, but we prefer to retain the current formulation as it provides a clearer differentiation.
>
> ---
>
>
> **Weakness 4:  Omit an explicit case split on Boundary dissimilarity for balls**
>
> **Response:**
> In the ball case, the two explicit cases can be represented using the same formula, so an explicit case split is not necessary. We have added this clarification in the new version of the paper to avoid confusion (in blue at Line 307).
>
> ---
>
> **Question 1:  Prove results by classical isometries between hyperbolic spaces**
>
> **Response:**
> Thank you for the suggestion. If I understand correctly, you are proposing that we use the isometry in Proposition 1 to derive our main results. Unfortunately, this approach does not apply in our setting. Our results concern arbitrary functions $f$ and $g$, which need not preserve the geometric structure of hyperbolic space or the conclusions that rely on it. In particular, even fundamental geometric properties of hyperbolic space—such as the triangle inequality—do not necessarily hold for our depth-dissimilarity. For this reason, the classical isometric framework cannot be used to establish our results.

---

> > ### Comment · Reviewer_ki2e · 2025-11-20
> >
> > Thank you for clarifying my question and for the general answer.
> > I will mantain my score.

---

### Official Review · Reviewer_RDte · 2025-11-01

**Soundness:** 3
**Presentation:** 2
**Contribution:** 2
**Rating:** 4
**Confidence:** 4

**Summary:**

This paper presents a novel hierarchical embedding method, which uses depth-dissimilarity and boundary-dissimilarity to assimilate Euclidean spaces to hyperbolic spaces. The authors proved that, by choosing appropriate depth-dissimilarity functions, the method equals traditional hyperbolic embedding models. In other words, the method is a generalization of hyperbolic embedding to allow tailored hierarchy embedding.

**Strengths:**

1. The authors generalize hyperbolic embedding models by assigning depth-dissimilarity and boundary-dissimilarity functions to Euclidean spaces. It allows sub-exponential distance scaling to adapt to richer structures than hierarchy.

2. The authors connect non-linear distance metrics with ball/box representations. It allows more expressiveness while maintaining the semantic modeling capabilities needed for ontology.

3. Clear mathematical derivations and proofs.

**Weaknesses:**

1. Citation format errors: many of the citations should be in brackets.

2. The authors motivated the choice of a Euclidean alternative to the traditional hyperbolic embedding by saying "However, as shown in Table 1, hyperbolic methods often rely on specialized constructed objects as embedding candidates, limiting their generalizability to data that encodes richer semantics beyond hierarchy" (Line 51-53). This is a very vague, descriptive motivation that may confuse readers -- does a specialized constructed object necessarily mean bad generalizability to non-hierarchy structures? Is there any theoretical support in literature for your motivation?

You may consider referring to this paper (https://arxiv.org/pdf/2407.16641?) -- the "curvature" of a subspace you need to sufficiently embed a certain size of hierarchy relies on the *capacity* of the subspace. There is trade-off between capacity and numerical stability -- that is, it is beneficial to use a subspace that has just adequate capacity. This can be a good support for your motivation -- your method effectively allows users to specify sub-exponential distance (i.e. between linear distance in Euclidean spaces and exponential distance in hyperbolic spaces) so that the capacity of subspaces match the sizes of hierarchical structures.

3. As Sala et al. (http://proceedings.mlr.press/v80/sala18a.html) discusses, equipping a coordinate space with exponential distance metrics (e.g. Poincare ball model or Lorentz model) poses an upper limit of numerical stability -- the reflective boundary. Due to the finite digits of float numbers, the depth of the hierarchy that can be losslessly embedded is limited -- i.e., the practical expressiveness of hyperbolic spaces is not infinite as the theory permits. What is the limit of numerical stability of the proposed Euclidean space alternative? From how I understand it, there is trade-off between how fast the depth-dissimilarity increases and how deep a hierarchy the method can stably embed. What is the trade-off for your method?

4. Equation 7 and 8 lack theoretical motivation. Why can and should the two dissimilarities be weighted summed? Is there any convex guarantee?

**Questions:**

Can the authors provide more insights on Weakness 3 and Weakness 4?

What are the "richer" structures that your method applies better to than simple hyperbolic embedding?

---

> ### Author Response · Authors · 2025-11-19
>
> Thank you for your review and feedback. We respond to your comments and questions below.
>
> ---
>
> **Weakness 1: Citation format errors**
>
> **Response:**
> These issues appear to arise from browser-based PDF rendering. The citations display correctly when the PDF is downloaded and opened locally. We kindly ask the reviewer to check the downloaded version, and we are happy to revise further if the problem persists.
>
> ---
>
> **Weakness 2: Unclear Motivation**
>
> **Response:**
>
> We appreciate the reviewer’s comment and will clarify our motivation more explicitly. When we refer to *“limiting generalizability for richer semantics beyond hierarchy,”* we mean that specific geometric representations used in prior work do not naturally support additional semantic operations. For example, cones in EntailmentCones and ShadowCones are not closed under intersection and therefore cannot represent conjunctions. We have added this clarification to the revised paper (Line 53).
>
> Regarding representational capacity, we thank the reviewer for the suggestion, but note that the concern is less applicable to **region-based methods** such as EntailmentCones and our RegD framework. These models generally have **infinite capacity**: a node represented as a region usually admits an unbounded number of subregions as potential children, with pairwise distances that can grow arbitrarily large (stated in Theorem 1, Item 2).
>
> In contrast, the works mentioned by the reviewers, such as Sala et al. (2018), focus on **point-based** representations and assume that all child embeddings lie on a sphere centered at the parent. This creates an inherent capacity limit: only finitely many points can be placed on a sphere while maintaining pairwise distances above a given threshold.
>
>
>
> ---
>
> **Weakness 3: Numerical Stability**
>
> **Response:**
> We thank the reviewer for raising the reflective-boundary issue in hyperbolic embeddings. As discussed in Section 3.2 of Sala et al. (2018), embedding a point at hyperbolic distance $D$ from the origin requires roughly $D$ digits of precision—far more than the $O(\log D)$ digits needed in Euclidean space—thus imposing a practical limit on the depth that can be stably represented.
>
> In our framework, **numerical stability is determined by the choice of the function $g$** in Definition 1, which controls how fast the depth-dissimilarity grows. If $g(x) = \text{arcosh}(1 + x)$, as in Proposition 1 where our distance becomes equivalent to the hyperbolic metric, the same instability arises.
>
> However, when $g$ is **linear**, as in Proposition 2 and in all our implementations, this issue is avoided. For example, in Equation (2), placing $\mathrm{Ball}(\mathbf{c}, r)$ at depth-dissimilarity $N$ from $\mathrm{Ball}(\mathbf{c}_0, r_0)$ requires:
> $$
> d = \frac{||\mathbf{c} - \mathbf{c}_0||_p^p + |r - r_0|^p}{r \, r_0}.
> $$
> Thus, representing $\mathbf{c}$ and $r$ requires only $O(\log d)$ digits.
>
>
> ---
>
>  **Weakness 4: Motivation of Equations 7 and 8**
>
>  **Response:**
> The energy function Equation (7) follows the previous work of Poincaré embedding, using a **symmetric–asymmetric decomposition** to model hierarchical relations:
>
> - Symmetric component (depth dissimilarity): Encourages proximity between semantically related nodes regardless of hierarchy direction.
> - Asymmetric component (boundary dissimilarity): Encodes directional, parent–child relationships.
>
> The loss function in Equation (8) is the same as the one used in ShadowCones, which refines the learning dynamics by adopting a contrastive-style loss and has been shown to achieve better performance according to their experiments.
>
>
> ---
>
> **Question 1: Weakness 3 and 4**
>
> **Response:**
> Addressed in the sections above.
>
> ---
>
>  **Question 2: Richer structures beyond hierarchy**
>
> **Response:**
> In this paper, we mainly take $\mathcal{EL}$-ontologies as examples. They include **conjunctions** ($\sqcap$) and **existential qualifications** ($\exists r.$) , which that go beyond simple subclass hierarchies ($\sqsubseteq$).
> Our framework supports arbitrary region representations  and thus can be integrated with existing ontology embedding methods.
> In contrast, standard hyperbolic embeddings — such as EntailmentCones, which use specific cones — cannot be directly extended to capture such semantics.

---

> ### Author Response · Authors · 2025-11-27
>
> Thank you again for your review and feedback. We hope the above clarifications have addressed your concerns. If you have any remaining questions, please don’t hesitate to let us know—we’re happy to address them.

---

### Official Review · Reviewer_octz · 2025-11-03

**Soundness:** 3
**Presentation:** 3
**Contribution:** 3
**Rating:** 6
**Confidence:** 3

**Summary:**

The paper introduces RegD, a general framework for embedding hierarchical and ontological structures using Euclidean region embeddings with a novel energy function. It combines boundary dissimilarity and depth dissimilarity to approximate hyperbolic properties like transitivity and layer separation. The method supports various geometric forms (e.g., boxes, balls) and integrates into existing ontology embedding frameworks. RegD achieves state-of-the-art results on multiple hierarchy and ontology benchmarks.

**Strengths:**

Flexible & Generalizable: Works with different region types (boxes, balls) and integrates into ontology models like ELBE and ELEM.

Hyperbolic-Like Expressiveness in Euclidean Space: Captures hierarchy depth and transitivity via depth dissimilarity without requiring hyperbolic geometry.

Strong Empirical Performance: Achieves state-of-the-art F1 scores on both DAG inference and ontology reasoning tasks.

**Weaknesses:**

Explain the further application of such method. Would you be able to use it in conjunction with a Neural Encoder derived from LLM?

**Questions:**

N/A

---

> ### Author Response · Authors · 2025-11-19
>
> Thank you for your review and feedback. We respond to your comment below.
>
>
> ---
>
> **Weakness: Details for applications**
>
> **Response:**
> Regarding applications, as noted in Lines 483–484, integrating our method with neural encoders based on large language models (LLMs), such as the approach in [1], represents a promising avenue for future work. Specifically, [1] introduced a framework that combines LLMs with Poincaré embeddings to model hierarchical structures from textual information. More recently, [2] extended this framework from atomic concepts to complex $\mathcal{EL}$-concepts through verbalization. However, due to inherent limitations of Poincaré embeddings, certain logical operators—particularly conjunction—remain difficult to model effectively. Our approach has the potential to overcome these shortcomings by using boxes as embedding regions while maintaining expressivity comparable to that of hyperbolic spaces.
>
> We have added additional details—highlighted in blue—clarifying this potential application in Section 5.
>
> [1] He, Yuan, et al. "Language models as hierarchy encoders." *Advances in Neural Information Processing Systems* 37 (2024): 14690–14711.
>
> [2] Yang, Hui, et al. "Language Models as Ontology Encoders." International Semantic Web Conference. Cham: Springer Nature Switzerland, 2025

---

### Author Response · Authors · 2025-12-02

Dear AC, PC, SPC, and Reviewers,

Thank you for taking over our submission. Below is a brief overview of the review situation.

## Main Strengths
The reviewers have explicitly mentioned the following strengths in the strengths section of the review:
- Flexible and general Euclidean region framework (**Reviewers octz, RDte, ki2e**).
- Clear and well-structured theories (**Reviewers RDte, ki2e, Lt9K**).
- Strong and consistent performance (**Reviewers octz, Lt9K**).

## Reviewer Concerns and Responses

| Reviewer | Main Concerns | Our Response | Status (empty for no response) | Reviewer Quote |
|----------|---------|--------------|--------|----------------|
| **octz** | Details on further applications, e.g., integration with LLMs | Provided detailed explanations | | |
| **RDte** | Motivation for Euclidean regions vs. hyperbolic embedding | Emphasized benefits of arbitrary Euclidean regions supporting semantic operations such as conjunction | | |
|          | Numerical stability problem of hyperbolic embeddings in our case | Stability determined by choice of *g*; no extra digits needed for hyperbolic embeddings in our implementation | | |
|          | Motivation for score function structure | Score function follows symmetric–asymmetric decomposition from classic work | | |
| **ki2e** | Narrowness of some theoretical results | Extended theoretical results to arbitrary regions | ✓ Resolved (confirmed Nov 20) | "Thank you for clarifying my question and for the general answer. I will maintain my score." |
|          | Empirical study of different region parameterizations | Empirical study provided | ✓ Resolved (confirmed Nov 20) | |
| **Lt9K** | Theoretical results limited to box/ball cases | Extended theoretical results to arbitrary regions | ✓ Resolved (confirmed Nov 25) |   "Most of my concerns have been addressed, but there are still two points that I am unsure about."  See **Lt9K** (2nd turn)|
|          | Missing distance-based metrics | Clarified focus on inferred edges and ranking metrics | ✓ Resolved (confirmed Nov 25) | |
|          | Simplicity of method vs. hyperbolic methods | Explained simplicity; added missing assumption and HNN extensions | ✓ Resolved (confirmed Nov 25) | |
| **Lt9K** (2nd turn) | Missing assumption on the region space | Added missing assumptions | | |
|          | Details on the possible extension on HNN | Provided more details | | |

Respectfully,
The Authors

---

### Meta-Review · Area_Chair_vxQC · 2025-12-14

**Summary:**

This paper proposes to embed data into Euclidean regions and their implicit hyperbolic structure. The reviewers highlight that the authors' approach is clear and recovers key hyperbolic properties, achieving empirical success on DAG and ontology datasets. The reviewers are concerned on the claim of arbitrary parameterized regions, as the method is mostly tested on balls/boxes, and the motivations of the loss and using Euclidean regions. The authors respond by adding theoretical extension in A.5, and through analogy to prior work (ShadowCones, etc.) and through emphasizing disadvantages of hyperbolic embedding. These weaknesses would require a major revision to address, which is not provided in the rebuttal and revision. Therefore, I recommend rejection. Furthermore, embedding data as probability distributions which are soft-regions (e.g. Gaussian manifold is hyperbolic) has been used in prior work, which the authors should connect to.

**Reviewer Concerns:**

Reviewer Lt9K is concerned on claim of arbitrary parameterized regions, which, after rebuttal, is still be outstanding, as it requires rework of the claims and theoretical results.

Reviewer RDte is concerned on the motivation, which requires principled justification and re-work of the early sections.

**Reviewer Scores:**

I appreciate the authors' summary of the rebuttal. However I could not identify strong evidence that the critic reviewers would increase their scores.

---

### Decision · Program_Chairs · 2026-01-26

Reject